# Modelling Health Financing Performance in Europe in the Context of Macroeconomic Uncertainties

**Marius Sorin Dincă** [1] , **Valentin Marian Antohi** [1,2,*] , **Maria Letiția Andronic** [1] , **Monica Răileanu Szeles** [1] and **Camelia Mirela Baba** [1]

1   Department of Finance, Accounting and Economic Theory, Transilvania University, 500036 Brasov, Romania;
    marius.dinca@unitbv.ro (M.S.D.); letitia.andronic@unitbv.ro (M.L.A.); monica.szeles@unitbv.ro (M.R.S.);
    mirela.baba@unitbv.ro (C.M.B.)
2   Department of Business Administration, Dunarea de Jos University, 800008 Galati, Romania
*   Correspondence: valentin.antohi@ugal.ro

**Abstract:** This paper makes a comparison between the financing of health systems in six European Member States: France, Denmark, Spain, Bulgaria, Romania and Hungary, starting from the structure of financial allocations to health systems in the context of fluctuating macroeconomic developments marked by multiple economic crises and the onset of the pandemic, which posed a real challenge to maintaining the health security of the European population and beyond. The need for this research is connected to the gap in the literature regarding economic development, health management and health financing performance. The main objective of the research is to determine the performance aspects of health systems financing and efficient financing models in relation to the evolution of macroeconomic indicators such as gross domestic product, household final consumption, general public expenditure and population. Empirical and analytical methods consisting of literature review, database construction, econometric modeling and statistical model validation were used. The results of the study highlight the performance of financial allocations for the six countries analyzed and could help decision-makers adjust health financing strategies in line with the insights provided by the current research. The novelty of this research is the comparison between different EU member states according to their economic development level in direct connection with health financing performance. This paper identifies the key aspects of health systems' financing and of efficient financing models in connection to the evolution of main macroeconomic indicators.

**Keywords:** health systems; financial performance; financial allocations; econometric modeling; macroeconomic developments

## 1. Introduction

Health is a fundamental human right, as stated in the 1948 World Health Organization Constitution (World Health Organization 1948), and universal health coverage is essential for the fulfillment of this right (World Health Organization and World Bank 2022). Achieving universal health coverage is one of the United Nations' (UN) important health-related sustainable development goals when the world's nations adopted the Sustainable Development Goals in 2015 (United Nations 2015). Thus, universal health coverage aims to provide quality health services for all without financial hardship, as outlined in the UN 2030 Agenda for Sustainable Development. This is essential for social inclusion, gender equality, poverty eradication, economic growth and human dignity, as reaffirmed in the Political Declaration of the UN Summit on Universal Health Coverage in 2019 (United Nations 2019). Universal health coverage ensures that all people have access to quality health services regardless of financial constraints, covering the whole life cycle from health promotion to prevention, treatment, rehabilitation and palliative care, without being a financial burden (World Health Organization 2023b).

Together with the World Health Organization (WHO) and the World Bank (WB), the Organization for Economic Co-operation and Development (OECD) is working to strengthen the universal health coverage strategy (World Health Organization et al. 2023) and to engage stakeholders to ensure progress.

OECD countries provide affordable health services yet are facing challenges in sustaining and improving these systems (OECD 2023). These issues are even more relevant in low- and middle-income countries, highlighting the need to expand coverage and improve health outcomes for all.

The persistence of social inequalities remains a significant obstacle for achieving universal health coverage. Even where national progress includes health coverage, the overall statistics hide the presence of disparities within countries. Due to the financial difficulties of people living in poor families, according to WHO (World Health Organization 2023b), monitoring health inequalities is significant to identify and monitor the segments of population that are disadvantaged. This information is needed to provide policy makers with a solid evidence base, which can then be used to develop policies, programs and practices that prioritize equality and achieve universal health coverage in a progressive way. There is a need for good quality data, improved statistics on gender disparities, socioeconomic disadvantages, and the unique challenges faced by indigenous communities, refugees and migrants who have been displaced by conflicts, economic hardships and environmental disasters. There are also disparities between EU Member States, one example being the financial differences in treatment costs. Reducing disparities in the financing of treatment costs between EU Member States would adjust public health policies (Negoita et al. 2023).

Promoting and preserving health is essential for the well-being of individuals and for the long-term progress of economic and social systems. Recognition of this notion occurred more than three decades ago, as recognized by the signatories of the Declaration of Alma-Ata (World Health Organization 1978). These persons noted that achieving health for all would not only improve the overall quality of life, but also promote world peace and security.

There are many strategies to promote and maintain health, such as education, home ownership, nutrition, living conditions, employment and addressing the growing problem of ageing individuals, all having a significant impact on quality of life and mortality rates (Commission on Social Determinants of Health 2008). Addressing disparities in these areas will lead to a reduction in health inequalities and timely access to health services, which include a combination of activities such as promotion, prevention, treatment and rehabilitation, is also essential. Achieving this goal is only feasible for a limited part of the population without a functioning and adequate health financing system. To ensure the affordability of health services, which is a key determinant of individuals' ability to access and use them as needed, member states of the WHO committed in 2005 to strengthen the development of national health financing systems to ensure universal access to services and mitigate the emergence of financial burden associated with health expenditures (World Health Organization 2005). Member States of the WHO have set themselves the goal of developing their health financing systems to ensure that all people can use health services, finding solutions for ensuring the access to these services even for the less financially capable persons.

A WHO report (World Health Organization 2010) outlined the steps countries can take to reform their financing structures, thereby accelerating progress towards the goal of universal coverage, while ensuring the long-term sustainability of the process. It is important that financing systems are designed to provide access for all people to quality and affordable health services, without exposing the patient to financial hardships. Drawing on the experience of several countries, the paper presents a comprehensive plan of action for nations at different levels of development and suggests strategies to increase global assistance to low-income countries in their quest of achieving universal healthcare coverage

and improving health outcomes. Health financing is becoming an important part of wider efforts to ensure population's social protection.

Examining health care's financial equity involves an analysis of health care services in relation to the demands of individuals and their financial ability to afford such services. An equitable health care system is characterized by the provision of equal services to individuals from different social and economic strata, with the expectation of equal remuneration for these services (Rostampour and Nosratnejad 2020).

Financial resources play an important role in supporting health services' delivery. In health systems, these resources are indispensable for purchasing medicines, health supplies, reagents and consumables. In addition, they are essential for maintaining the infrastructure and operations of healthcare institutions, as well as for compensating health professionals for their services. However, it is important to recognize that financial resources are limited and serve as a pervasive limitation that is shared by health systems everywhere. Health financing is recognized by the WHO as a significant component of the six key determinants of health systems. It emphasizes that sufficient funding is important for the proper functioning of the remaining five parts (World Health Organization 2010). To achieve the overall goal of universal health coverage worldwide, most financing schemes have undergone adjustments in the context of significant economic challenges, substantial economic constraints, political instability and inadequate governance. Many countries have limited fiscal capacity, inadequate formal social protection mechanisms for marginalized populations and a lack of government oversight in the informal health sector. Addressing these challenges requires joint public sector and community participation in health care financing to increase health care's access for the poor and provide social safeguards against health care spending.

The main objective of public health systems is to allocate resources for disease prevention and to provide financial guarantees for health care. No nation can claim to have made substantial progress towards achieving universal health coverage without relying heavily on public funding for healthcare (Barroy et al. 2017; Kutzin 2013).

The health financing policy approach places the sector in the context of broader public budgeting systems and emphasizes the essential role of the state budget in facilitating universal health care, as well as the imperative to generate additional revenue and explore new financing avenues. A comprehensive understanding of the fundamentals of public budgeting is necessary for those with an interest in health financing (Barroy et al. 2018; Maeda et al. 2014).

Improving universal health coverage requires greater involvement of governments around the world in health care budgeting. This can be achieved by raising awareness of the importance of public health budgeting through the implementation of specialized budget classifications. In addition, it is essential to establish a strong and resilient public budgeting framework supported by well-funded programs. Finally, it is imperative to incorporate effective budgeting practices to address the challenges facing the health sector.

An important challenge has been the implementation of fiscal policy measures by governments around the world in mitigating the adverse effects of the COVID-19 pandemic on poverty levels. Public spending in rich countries has demonstrated significant effectiveness in alleviating poverty, successfully offsetting the adverse effects of the COVID-19 pandemic through the implementation of revised fiscal policies and various emergency support programs. The budgetary consequences of the COVID-19 pandemic have put additional pressure on nations' financial resources, affecting public funding, which is crucial for sustaining health systems. This has had implications both for ensuring health security and for the efforts of achieving universal health coverage. The COVID-19 pandemic has had a worrying effect on health budgeting procedures, as the imperative to maintain high levels of public spending has not been directly aligned with full coverage of all health needs. Globally, the COVID-19 pandemic has had several impacts on different dimensions, including the global economy (Chattu et al. 2022). At the peak of the COVID-19 epidemic in 2020–2021, the global community faced an unprecedented level of social and economic

consequences not seen in contemporary times (Li et al. 2023) and led to a catastrophic burden on health systems and increased spending on supporting health infrastructure (Vysochyna et al. 2023). The global epidemic has profoundly impacted the financial stability of hospitals, health systems, nations and international organizations, resulting in substantial economic losses.

The global economy is currently facing a multitude of challenges. The current economic landscape is characterized by high levels of inflation, exceeding those from previous decades. This is accompanied by the significant financial constraints in many regions, which, coupled with Russia's invasion of Ukraine and the persistency of the COVID-19 pandemic, creates a very uncertain global economic outlook, with many economies experiencing a slowdown of economic growth or even recession.

The efficient management of health budgets requires a change in spending behavior, which is a challenging aspect of budget execution. Service providers are working to establish a genuine financing system based on tangible results. More attention needs to be paid to the link between budget formulation and providers' payment reforms. It is essential to assess the extent to which these processes align and reinforce each other, as they play a crucial role in determining health outcomes.

The papers started from the premise of considering two opposite groups of countries, i.e., France, Denmark and Spain occupy the first three positions at the level of the EU in terms of economic development, whilst the remaining three countries, respectively Bulgaria, Romania and Hungary, have common economic characteristics like geographic position (neighboring states), have joined the EU in the same period, being part of the new Member States (Hungary in 2004, Bulgaria and Romania in 2007), were under communism and shortly after the fall of communism had adopted similar health systems. In this context, we analyze general government financing schemes and financing schemes with compulsory contributions to the health system (HF. 1) as well as out-of-pocket payments by the population for health care (HF. 3) for a number of six European Union (EU) member states, as they account for the most significant share of revenue and therefore of expenditure in health systems.

The six EU member states for the study were chosen based on the European Health Care Index ranking by country in 2023 produced by the online database (Numbeo 2023). The Health Care Index is a comparative tool that assesses the overall quality of a health system, including factors such as health professionals, equipment, staff, doctors and costs. The selected countries are differently economically developed countries, with unique features, characterized by certain types of health systems, as seen further in Section 3.2.

The authors propose the following research questions:

Q1. Is health performance directly dependent on the allocation of government financial funds?

Q2. Does the public debt the performance of the financing health system?

Q3. Is performance in health systems dependent on the level of final consumption expenditure of households of a state? Q4. Can a financial allocation model be defined in order to identify vulnerabilities of health systems in achieving a new level of performance?

In order to response to these questions, we establish the following *objectives*:

O1. The study of specialized literature.

O2. Presentation of the financing mechanisms, highlighting the health care index in the six EU selected countries.

O3. Conceptualization of a financial model of efficient allocation in an uncertain macroeconomic context.

O4. Econometric model testing and validation.

O5. Dissemination of results.

The justification of the research consists in identifying a new research domain regarding the connection between economic development, health management and health financing performance. As a result, the findings of this research are notable as well.

## 2. Literature Review

Access to healthcare is an important issue for everyone and economic status influences this. The impact and importance of the healthcare sector in the economy needs to be analyzed as healthcare becomes more complex and its associated expenses increase. The financing of the health sector influences the health status of the population, an aspect discussed by many authors over a large number of publications dedicated to this field. To analyze them, a synthetic review was performed in accordance with the general principles of literature review. The Web of Science database was used to search for the term "healthcare financing" used in the titles, abstracts, and keywords of the publications included in this database. For the 2020–2023 period, a total number of 3,958 articles identified on the Web of Science platform were acknowledged in 24,232 specialist papers that cited them (without self-citations), with an average citation rate of 6.95 per article and a Hirsch index of 49 points (Figure 1).

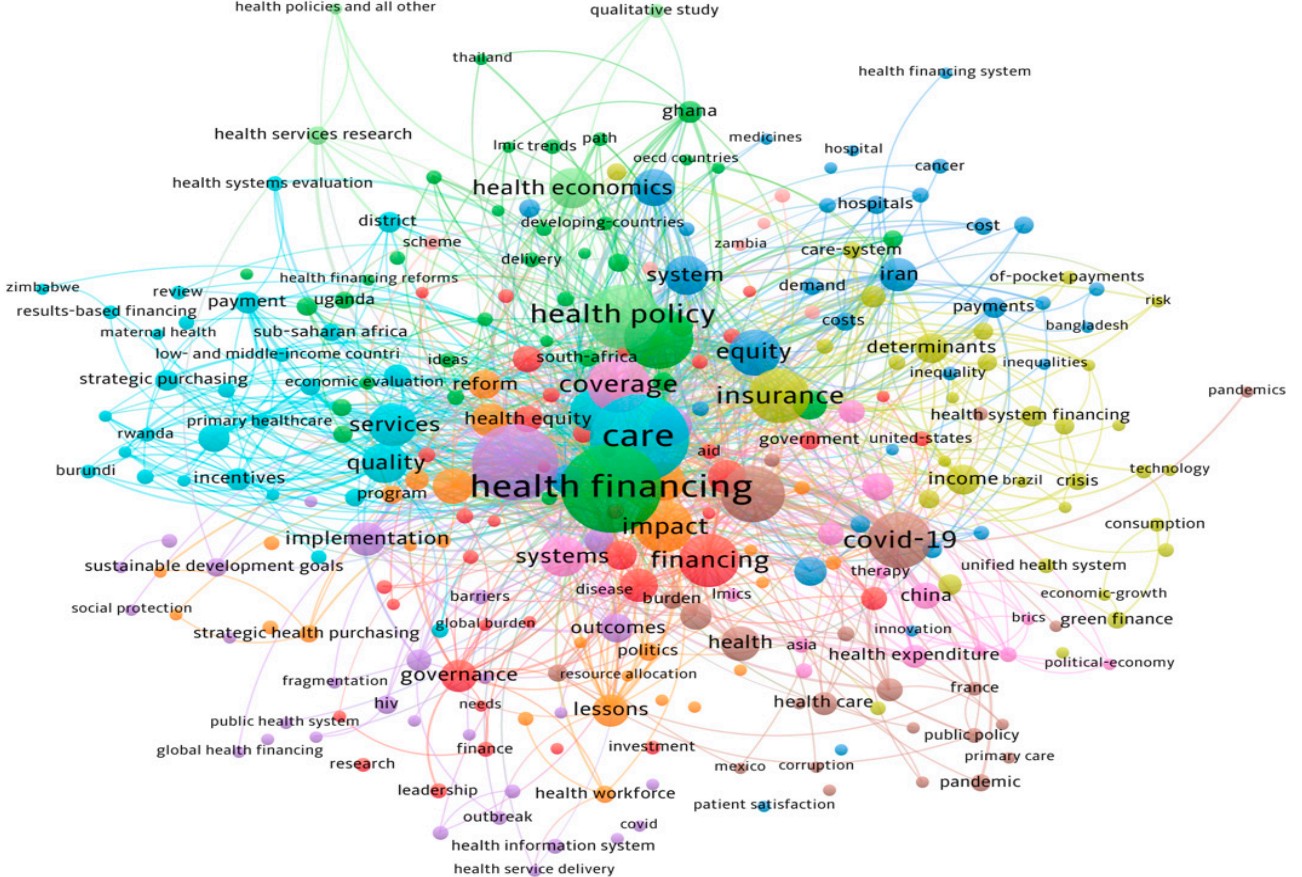

**Figure 1.** Bibliometric analysis regarding the financing of the health system. Source: the authors using VOSviewer software (https://www.vosviewer.com/, accessed on 10 October 2023).

An interesting analysis by a group of authors (Jagrič et al. 2021) examined the economic importance of the healthcare sector in 19 European national economies, using input–output tables and sectoral data. The results show that increased spending on health products and services has positive effects on national economies, particularly in terms of value added, employment and household income. The importance of the health sector is linked to countries' levels of development, with the benefits being particularly promising in countries with lower GDP per capita. Changes in the healthcare sector can play a significant role in economic policy.

In another study, the authors (Gabani et al. 2023) analyzed the impact of health financing mechanisms on health system outcomes in low- and middle-income countries

and suggested that government-funded systems are better in improving most outcomes compared to the system of health insurances. The study found that transitions to government funding increased life expectancy, reduced under 5 years' mortality, and reduced the incidence of catastrophic health spending. However, increased non-contributive government funding, rather than social health insurances, is not likely to improve health system outcomes due to higher implementation costs and limited coverage. This could be a warning to policy makers considering reforms of social health insurance systems to achieve universal health coverage.

The disparity between increasing health care coverage rates and the lack of progress in health outcomes in low- and middle-income countries underscores the need to allocate resources to improving the quality of health care in addition to ensuring accessibility. In a recent research, Piatti-Fuenfkirchen et al. (2021) argued for the need for a reform of the financing of health systems known as performance-based financing (PBF). PBF refers to the allocation of financial resources to healthcare institutions, asking them to provide a pre-defined range of services in accordance with established quality and management criteria. These initiatives have started a comprehensive range of reforms, including provider autonomy, increased access to financial services, increased flexibility in the use of funds, a performance-oriented approach to budget allocation, and the implementation of standards of rigorous verification.

Another research (de Walque and Kandpal 2022) indicates that in under-financed—centralized health systems, PBF can lead to improvements in the use of services. However, its effects on quality have been shown to be limited. Therefore, it can be argued that pay for performance, which is the main element of performance-based financing, does not appear to contribute much more to benefits compared to flexible payment systems and provider autonomy. To improve the quality of health care delivery, it is recommended that health financing strategies move away from performance-based remuneration while maintaining the key components of direct facility financing, autonomy, transparency and community involvement.

Macroeconomic, public and private health insurance financing factors have a significant impact on health care expenditures made from direct payments of the population. Research conducted by some authors (Grigorakis et al. 2018) for 26 EU and OECD countries for the 1995 to 2013 period found that public and private financing have a significant offsetting effect on direct payments expenditure, while unemployment rates have a positive impact. The findings suggest that policymakers should consider public and private funding institutions, as they have an inverse effect on spendings from the direct payments. The study suggests that countries should provide citizens with financial risk protection against direct payments with effects on public finances. Access to healthcare is very important for everyone, and the economic status of a country significantly influences this aspect. As health care becomes more complex, increasing spending requires a better understanding of health sector's impact on the country's economy. Mihalache and Apetroi (2020) analyzed the efficiency of public health's financing system in Romania, with an emphasis on population's health and the share of financing influenced by the hospitals' performance, technology and human resources. The paper aims to quantify the degree of efficiency in the financing of health services in Romania in the European context using Data Envelopment Analysis (DEA) and Euro Health Consumer Index (EHCI), providing a comprehensive overview of the financing of health services in Romania.

Dincă et al. used DEA to analyze the European health system financing schemes, according to countries' affiliation to either the Beveridge or the Bismarck system. Their research used five input variables describing the financial and human resources, the level of health infrastructure, the medical technology and the healthcare utilization and four output measures expressing the overall health status of the population and the effectiveness of prevention and emergency care. The authors found that the most efficient healthcare systems were in Sweden, the UK and Romania (Dincă et al. 2020).

Another group of authors (Petre et al. 2023) carried out an analysis of the health system in Romania, focusing on its strengths, weaknesses and the impact they have on access to

quality services. The system faces challenges such as insufficient funding, lack of medical personnel and inefficient service delivery, which hinders accessibility, especially in rural areas. The research presents three hypotheses, respectively, inadequate funding has a negative impact on health facilities, insufficient medical staff contributes to inequalities in access and provision, and inadequacies in service delivery prevent timely and effective delivery of health care. The analysis includes a comprehensive analysis of key aspects of the Romanian healthcare system, including infrastructure, financing mechanisms, service delivery and outcomes. Romania has a mixed health care system, with family doctors providing primary care and hospitals and specialized medical centers providing secondary and tertiary care. The study underscores the need for significant change to address these issues and achieve unbiased and affordable healthcare.

All healthcare systems are facing rising costs and increased demand. One of the best health systems in Europe is facing a similar situation, namely the German one, where the increase in costs of medical services, lead political decision-makers to focus on modern infrastructure and high-quality services, with a system of preventive, interconnected and modern health. Two authors (Schmitt and Haarmann 2023) proposed to interpret the German government's plans and determine how Germany can finance prevention programs, health promotion measures and innovative solutions demanded by the rising medical care costs. Through the study, the authors suggest the establishment of a prevention fund and a flexible remuneration model for digitalized care. This could help identify sustainable funding approaches for health promotion, prevention and innovation in the German healthcare system.

Health financing is a fundamental component of health systems, comprising three interrelated functions, respectively revenue mobilization, pooling of funds and procurement of health services. Other authors (Evans et al. 2023) carried out a comprehensive analysis of four fundamental principles that are essential for achieving high performance in health financing across the three core activities. The first principle recommends that the primary source of funding for health care should be mandated pre-paid funds rather than individual out-of-pocket payments. The effectiveness of risk sharing is enhanced when insurance pools are large and encompass a diverse range of health risk profiles recommends the second principle stated by the authors. The third principle considers the procurement process, the identification of a fundamental package of healthcare services to be provided to all beneficiaries through common funds. The fourth principle maintains that the procurement process should use payment mechanisms that ensure the widespread availability of guaranteed high-quality services while minimizing costs. Other principles are anticipated to emerge in the future as health financing adapts to emerging and new risks. These threats may require increased spending on health systems or can hinder a country's ability to generate money.

An interesting research (Lo et al. 2019) discuss the importance of governments investing in health public goods to protect humanity's survival, such as improving air quality, developing sustainable food systems, preserving biodiversity and reducing greenhouse gas emissions. The authors find that environmental public spending does not exceed health public spending. They also suggest that improved governance and increased political participation have a positive impact on health outcomes by increasing environmental quality.

Effective risk adjustment is essential in health systems, especially in competitive health insurance systems. The provider's risk structure is critical to payment and performance, whereas optimal incentive-based service models depend on the provider's earnings level. Recently, health systems have been subject to the risk generated by the COVID-19 epidemic, which led to a major economic crisis, with a significant impact on individual and social well-being. Within health spending, a large percentage is allocated to curative care instead of prevention, whereas the impact of the COVID-19 pandemic on our society and economy has brought into focus the need to consider the resilience of health systems, the importance of health systems performance, alongside access to medical services and their quality and efficiency.

The COVID-19 pandemic triggered an economic shock ten years after the 2008 global financial crisis, testing health systems by reducing public revenues and increasing the need for publicly funded healthcare.

Thomson et al. (2022) analyzed the resilience of health financing policy to economic shocks in Europe, finding that some health systems were weakened by the 2008 crisis. Lessons learned from the 2008 crisis highlight weaknesses of health financing policy, especially in countries with health insurance systems. Building resilience requires reducing cyclicality in policy coverage, increasing the priority of health in the allocation of public spending, and ensuring that resources are used to meet equity and efficiency goals.

The above comprehensive analysis has been conducted on the economic significance of the EU healthcare systems. The findings reveal that augmenting investments in health-related goods and services has a beneficial effect on national economies, namely in terms of value added, employment and household income. Government-funded systems have been found to be more effective in enhancing health system outcomes when compared to health insurance. However, it is important to note that non-contributory government funding is unlikely to lead to improved outcomes due to the presence of higher implementation costs and limited coverage. Performance-based finance (PBF) has been suggested as a viable approach for enhancing healthcare quality, while its impact on quality improvement has been constrained. The healthcare expenses produced via direct payments are highly influenced by several aspects related to macroeconomic conditions, as well as the financing mechanisms of public and private health insurance. From the literature, we found that it is imperative for nations to offer their citizens financial risk mitigation in the form of safeguards against direct payments that may have adverse implications for state budgets.

The health system in Romania has various obstacles, including inadequate financial resources, a shortage of healthcare professionals and ineffective service provision. These issues have a detrimental impact on the accessibility of healthcare services, particularly in rural regions. Some of the above authors propose substantial modifications to effectively tackle these concerns and attain accessible healthcare. According to the literature review, health funding is an important element of health systems, encompassing four core principles that are important for attaining optimal performance. The allocation of government resources towards health public goods, such as the enhancement of air quality and the preservation of biodiversity, has the potential to yield favorable effects on health outcomes. The COVID-19 pandemic has underscored the imperative of building resilience within health systems and the significance of allocating public funds towards prioritizing health.

Even though the literature is huge, we consider that there is not enough research on the linkages between economic development, health management and health financing performance. In our opinion, the understanding of the relationship between economic development, health management and health financing performance is essential for policymakers and stakeholders. By filling this gap in the literature, we can gain valuable insights into reallocating public funds, overall economic growth and improved health outcomes. This research will allow building more resilient and more sustainable healthcare systems according to the identified vulnerabilities.

## 3. The Health Financing Systems

In every country, policy makers want timely and reliable data to develop health policies and oversee their implementation. Recognizing the importance of financing as a fundamental component of the health system, the integrity and accuracy of financial data is of paramount importance. Assessing the quality of health care, together with its financial support, is a significant issue, albeit difficult to quantify. The need for knowledge of health care financing and expenditure flows has led to the implementation of ways to estimate and monitor health financing in different countries at different stages to improve revenue generation and resource allocation strategies to reduce waste and ensure financial protection.

### 3.1. The System of Health Accounts

The need to cut waste and improve resource allocation has led international bodies to identify appropriate financing mechanisms at government level. To ensure this goal of measuring national health expenditure at the international level, the System of National Accounts has been implemented since 1953 and has been revised up to 2008 (United Nations 2010). Since 2000, through the efforts of WHO, Eurostat and OECD, a manual entitled System of Health Accounts (OECD et al. 2000) was created, which was revised afterwards in 2011 (OECD et al. 2011) and 2017 (OECD et al. 2017). The EU adopted the European System of Accounts, which was revised in 1970, 1995 and 2010 and became mandatory for all EU Member States (European Commission 2013). Accordingly, this manual establishes a set of statistical reporting rules and proposes a new International Classification of Health Accounts (ICHA), which covers three dimensions (Figure 2): health care by functions of care or types of health services, health care providers and sources or schemes of financing, thus ensuring increasing demand for basic financial data sets provided by National Health Accounts as well as for international comparisons of health care expenditure.

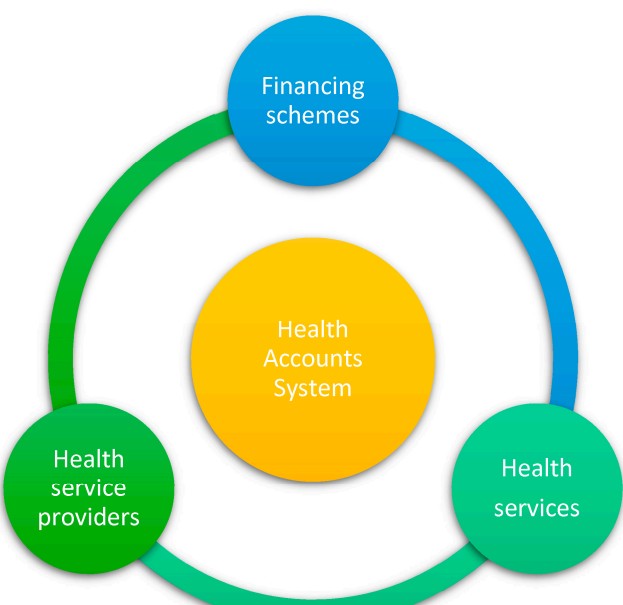

**Figure 2.** The triaxle relationship underlying the Health Accounts System. Source: Realized by authors based on information provided by the Health Accounts System 2011 (OECD et al. 2011).

The implementation of the System of Health Accounts has led to improved accountability, comparability and performance measurement, regardless of the wide variety of health system financing around the world. It has also led to increased standardization, consistency and policy relevance, while facilitating the continuous production of health accounts and their application in decision-making processes, achieving a stronger link between financing functions, i.e., revenue collection, bundling and allocation/purchase of services and accounting classifications. This has led to a more comprehensive presentation of the health financing system.

National health accounts thus serve as a valuable tool for evaluating health system reforms, as well as facilitating efficient resource allocation, by allowing analysis of health system performance (Rathe et al. 2018).

Data collection under the System of Health Accounts' manual by international bodies such as the WHO, the EU and the OECD began in 2016, and since 2017 EU Member States have been compelled to participate fully in its implementation.

Health financing is a fundamental element of health systems. The classification of health care based on financing can serve as a valuable tool for in-depth national assessments, helping health professionals to gain a comprehensive and transparent understanding of

health financing. When combined with the other two categories, respectively, the types of health services and health care providers, this tool provides the means to comprehensively represent both health care financing and the representation of the flow of financial resources within the health system. The classification includes parameters that can be compared within countries, between countries and in their time trends, providing valuable information for assessing the effectiveness of health system financing. Health financing systems are designed to generate and distribute financial resources within health systems and to meet the present and future health care needs of individuals and communities. The configuration of a financing scheme within a health system can be understood through two distinct components. Firstly, there are different financing schemes, including national health services, social health insurance and voluntary insurance. Secondly, there are institutional bodies that play a role in the financing process, such as various government institutions, social security agencies and private insurance companies. Health financing systems are structured around transactions that facilitate three primary health financing functions. These functions include the collection of revenue by governments through the payment of social security contributions to a unified national fund, the pooling and distribution of resources to funding bodies or agencies, and the purchase of health services from service providers. The System of Health Accounts introduced fundamental concepts aimed at describing the structure of health financing systems. These concepts focused mainly on the assessment of revenues generated by different types of health care financing systems, and the corresponding expenditures involved in the direct purchase of goods and services from health care providers.

A number of criteria were chosen to differentiate between the different health financing schemes, including the following: mode of participation and type of compulsory or voluntary contribution to the health system, residence or non-residence of participants in the health system, type of insurance payment, contributory or non-contributory, etc. The Health Accounts System considered classifying health schemes into five major or general categories (Figure 3).

Government funding schemes and mandatory health care financing schemes promote equitable and equal access to essential health care services for the whole population or certain groups, including vulnerable groups. In situations of uncertainty, social protection concerns become important and support measures need to be put in place to help vulnerable people and mitigate the associated social costs (Antohi et al. 2023). These schemes include a range of approaches to financing health systems, including government-funded programs, schemes designed to finance social health insurance through compulsory contributions, compulsory private insurance and compulsory health savings accounts. While government involvement in health systems plays a vital role in promoting equitable and equal access to essential health care services, no single funding scheme covers the full costs of health services; in addition to government funding schemes, there will always be direct payments from the population. Direct payments by the population for health care goods and services from their own income show the direct burden of health care costs on families when using health care services in low-income countries, and these payments are often the main form of financing health care, with catastrophic effects on the financial situation of many families. Three types of direct payments are distinguished, respectively, without cost-sharing, with cost-sharing or co-payment in government funding schemes and compulsory contributory health insurance schemes and with cost-sharing or co-payment in voluntary health payment schemes. Examining the weight assigned to each sub-category and the changes in their share over time provides a more comprehensive understanding of the financial burden imposed on families in terms of health financing, as well as the level of government intervention in health financing.

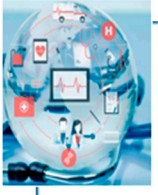
General government financing schemes and health-contribution financing schemes (HF. 1)

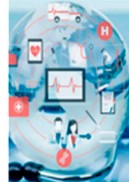
Voluntary health payment schemes (HF. 2)

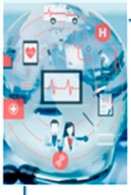
Household out-of-pocket payments for health care (HF. 3)

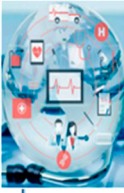
Funding schemes in the rest of the world (HF. 4)

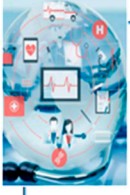
Other funding schemes not elsewhere classified (HF. 0)

**Figure 3.** International classification of general health funding schemes. Source: the authors based on information provided by the 2011 Health Accounts System (OECD et al. 2011).

*3.2. The Health Care Index and the Socio-Economic Context of the Selected Countries*

The health care index provides an assessment of healthcare infrastructure, services and resources in different cities or countries, taking an overall snapshot of healthcare in a given location.

According to the data provided by Eurostat (Eurostat 2023a, 2023b), it can be seen that health expenditures are allocated according to the level of country's economic development instead of population density (Figure 4).

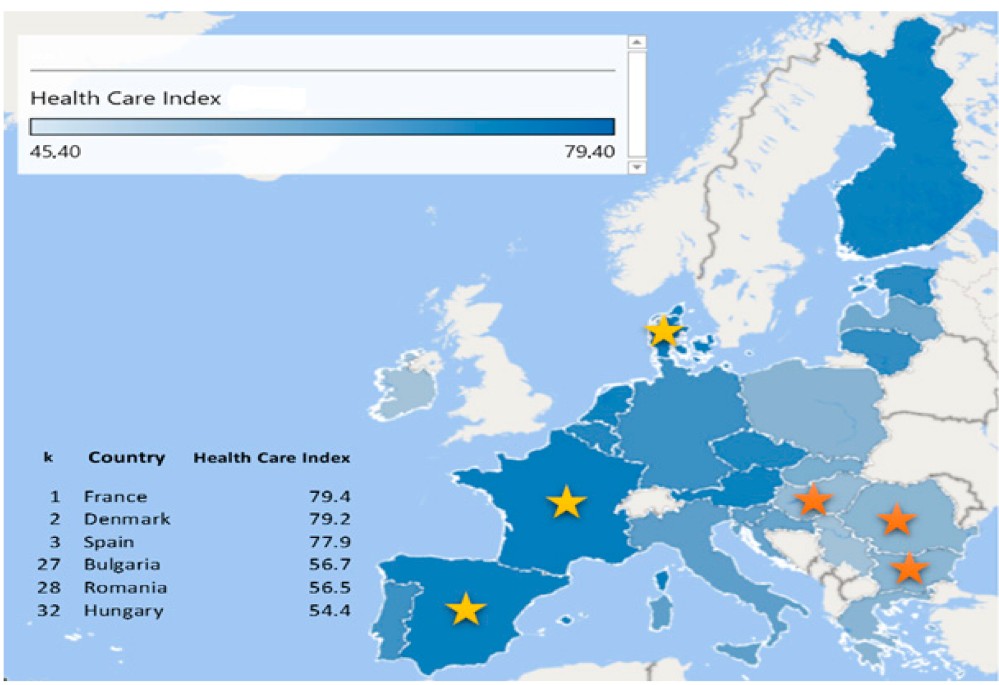

**Figure 4.** Ranking of European countries analyzed based on the Health Care Index in 2023. Source: the authors based on information from the Numbeo online database (Numbeo 2023).

Denmark has the highest allocation of overall health public expenditure as a percentage of GDP and the lowest number of inhabitants among the analyzed countries. At the same time, indicators such as the inflation rate clearly distinguish developed from less developed countries, with inflation rates nearly half as low as the ones from less developed countries. This can also be seen in the analysis of social welfare as reflected in GDP per capita (Figure 5).

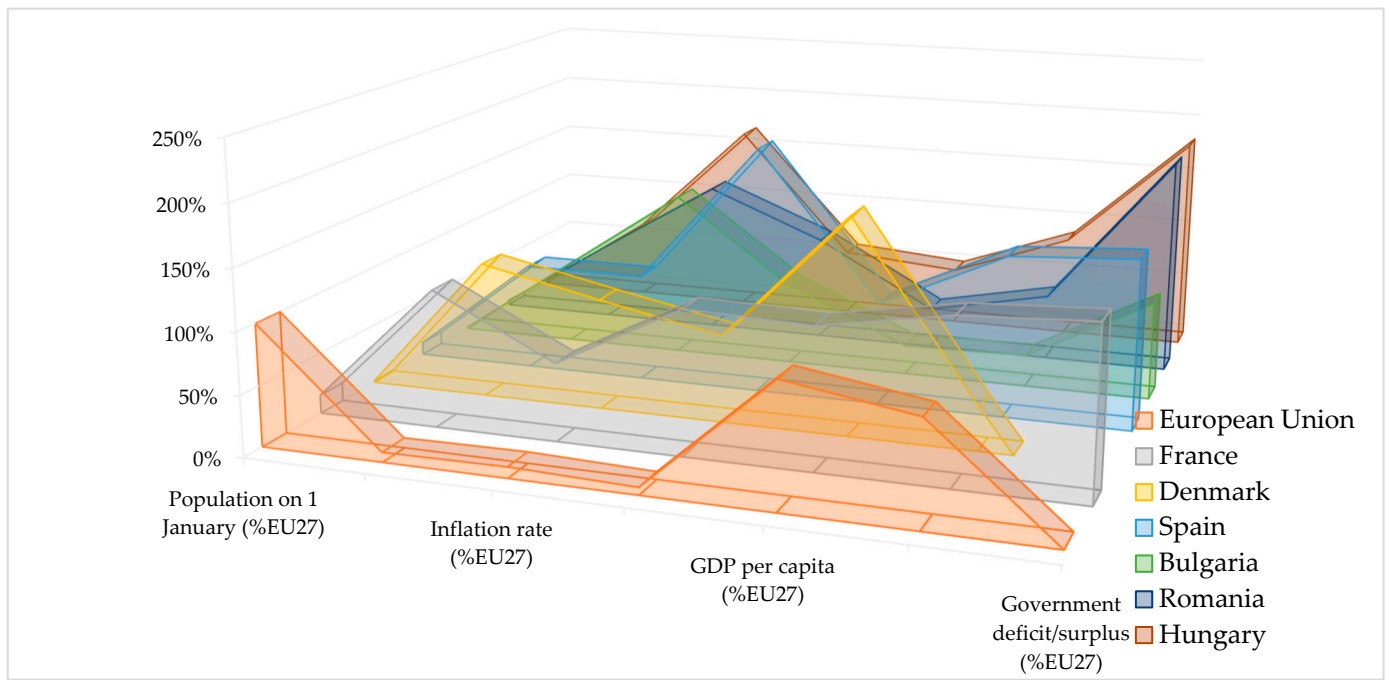

**Figure 5.** Dynamics of main socio-economic indicators compared to the EU average. Source: the authors according to information provided by Eurostat online database (Eurostat 2023a, 2023b).

The level of gross public debt in former communist countries is generally lower than in developed countries, as they have less experience with market economies than developed countries and as such the accumulation of public debt reflects this. In terms of budget deficits, with the exception of Denmark, all the other countries analyzed are exposed to budget deficits that generally exceed the 3% threshold of the GDP and reflect the state of uncertainty induced by the multiple crises in the European economies (Table 1).

**Table 1.** Socio-economic indicators for the analyzed countries.

| | Population on 1 January (2023) | General Government Expenditure on Health (% of GDP—2021) | Inflation Rate (% Change from Previous Year—2022) | Unemployment Rate (as % of Working Population Aged 15–74—2022) | GDP per Capita (Euro per Capita—2022) | General Government Gross debt (as % of GDP—2022) | Government Deficit/Surplus (% of GDP—2022) |
|---|---|---|---|---|---|---|---|
| European Union | 448,387,872 | 8.1% | 9.2% | 6.2% | 28,860€ | 84.0% | −3.4% |
| France | 68,070,697 | 9.2% | 5.9% | 7.3% | 33,180€ | 111.6% | −4.7% |
| Denmark | 5,932,654 | 9.2% | 8.5% | 4.5% | 51,660€ | 30.1% | 3.3% |
| Spain | 48,059,777 | 7.3% | 8.3% | 12.9% | 24,810€ | 113.2% | −4.8% |
| Bulgaria | 6,447,710 | 5.8% | 13.0% | 4.3% | 7250€ | 22.9% | −2.8% |
| Romania | 19,051,562 | 5.5% | 12.0% | 5.6% | 10,110€ | 47.3% | −6.2% |
| Hungary | 9,597,085 | 5.6% | 15.3% | 3.6% | 14,370€ | 73.3% | −6.2% |

France's health system incorporates a concept of social health insurance with a significant reliance on social health insurance contributions to finance health services. The health system offers comprehensive coverage to all individuals, encompassing a wide range of benefits. However, individuals are obliged to contribute financially by sharing the costs of the necessary services. The use of complementary private insurance to meet these costs leads to a significant reduction in average expenditure in direct payments (Or et al. 2023). The French health system is of a mixed type, including the Bismarck system (social health insurance), has universal social health insurance, and to ensure financial sustainability, health financing sources have been extended beyond compulsory social contributions to include a wider range of revenue sources, including active financial sources, earmarked investments and taxes and value added taxes. France's spending on health is among the highest in the EU at 9.2% of GDP in 2021, above the EU average.

The Danish health system is largely tax-funded, decentralized and organized on three administrative levels: state, regional and local. Planning and regulation take place at both central and local level. The national level is responsible for regulation, supervision, some planning and quality monitoring, while the five regions are responsible for defining and planning health service provision. Municipalities are responsible for health promotion, disease prevention, rehabilitation, home care and long-term care (OECD et al. 2021b). The Danish population is automatically insured under the national health system. Funding is predominantly provided from general tax revenues at the state level and, to a lesser extent, from a municipal income tax.

Spain's national health system is based on universal coverage and is mainly funded by social contributions and taxes. The Ministry of Health is responsible for national planning and regulation, health and primary care competences, resource allocation, procurement and delivery are transferred at regional level to the 17 regional health authorities. Financing of the health system is predominantly public, as for private financing, payments come from a combination of out-of-pocket payments and private health insurance (OECD et al. 2021d).

The Bulgarian health system is based on compulsory social health insurance contributions, with voluntary health insurance having a role. The National Health Insurance System of Bulgaria, through its regional health insurance subsidiaries in 28 regions, is the sole purchaser of health services. The Ministry of Health is responsible for the overall governance of the health system, the development of health legislation, the coordination and supervision of various subordinate bodies and the planning and regulation of health care providers. Even though its health expenditure per capita has increased constantly, Bulgaria still ranks last among EU countries, with direct payments still having a significant weight in health expenditure, (with a share of more than 38%) as can be seen from OECD report—Bulgaria's 2021 Country Profile on Health (OECD et al. 2021a).

The health system in Romania is based on the obligation of social health insurance contributions with a strong involvement of the state. The Ministry of Health is responsible for the general governance of the social health insurance system, while the National Health Insurance House administers and regulates the single national health social insurance fund, through the county directorates of public health and the county health insurance houses. Through the county health insurance companies, medical services are purchased from medical service providers, and the Ministry of Health ensures the payment of national health programs. Even if the social health insurance system in Romania is mandatory, a significant percentage (as stated in the OECD report—Romania's Country Profile of 2021) in terms of health (OECD et al. 2022) of approximately 11% of the population remains uninsured, especially in rural areas. Even if health spending has recently increased in Romania, it still remains one of the EU countries with the lowest health spending, both per capita and as a percentage of GDP.

Hungary has only one health insurance fund that provides medical coverage for the population. The fund is administered by the National Health Insurance Fund Management Institute, which operates under the direct control of the Ministry of Human Resources. The Ministry establishes development strategies, establishes financing conditions and the

package of medical benefits and has a regulatory role. Even though the health system expenditure is provided by public funds, there is still a high level of direct payments for medical services, of about 28%, as shown in the OECD report—Hungary Country Profile 2021 in terms of health (OECD et al. 2021c).

## 4. Research Methodology

The research methodology follows the following algorithm (see the chart in Figure 6).

| Start |
| --- |

| O1. The study of specialized literature. | | |
| --- | --- | --- |
| Presentation of bibliometric analysis | Researches about health financial approaches in literature review | Identyfing eventualy gaps in literature |

| O2. Presentation of the financing mechanisms, highlighting the health care index in the six EU selected countries | | |
| --- | --- | --- |
| Presenting of the System of Health Accounts | Presenting of the Health Care Index | Presenting of the actual Socio-Economic Context |

| O3. Conceptualization of a financial model of efficient allocation in an uncertain macroeconomic context. | | |
| --- | --- | --- |
| Indicators selection | Defyning working hypoyheses (H1–H6) | Conceptualizing financial model |

| O4. Econometric model testing and validation. |
| --- |

| O5. Dissemination of results. | |
| --- | --- |
| Formulating of conclusions | Formulating of public policies |

| Stop |
| --- |

| The stars signify the 6 states analysed. |
| --- |

**Figure 6.** Flowchart of analyses. Source: the authors.

To achieve the research objectives, public data made available by the WHO (World Health Organization 2023a), were used, analyzing the following indicators considered to be relevant following the literature research:

- CHE—Current health expenditure by financing schemes, in million current US$;
- GSCCHFS—Government schemes and compulsory contributory health care financing schemes, in million current US$;
- OOPS—Household out-of-pocket payments (OOPS), in million current US$;
- GDP—Gross Domestic Product (GDP), in million current US$;
- PFC—Final consumption expenditure of households and profit institutions serving households (PFC), in million current US$;
- GGE—General government expenditure (GGE), in million current US$;
- POP—Population, in thousands.

The analyzed data were studied for the 2000–2021 period (2021 being the last year of reporting), for six European states, respectively, France, Denmark, Spain, Bulgaria, Romania and Hungary. The countries were selected using the ranking made by the online database

([Numbeo 2023](#)) and refer to the states occupying the first three positions at the level of the EU, France, Denmark and Spain, whilst the remaining three countries, respectively Bulgaria, Romania and Hungary, have common economic characteristics like geographic position (neighboring states), have joined the EU in the same period, being part of the new member states (Hungary in 2004, Bulgaria and Romania in 2007), were under communism, and shortly after the fall of communism had adopted similar health systems.

Six *working hypotheses* were defined and will be tested within the research:

**H1.** *Health performance is directly dependent on the effective process of allocating government financial funds;*

**H2.** *The more the financing performance decreases, the more the level of direct financing by the population (household out-of-pocket payments—OOPS) increases;*

**H3.** *The more efficient the financing of health systems, the more it reduces the margin of amplitude of the minimum—maximum forecast interval of the financing function;*

**H4.** *Performance in health systems is directly dependent on the level of public debt of a state;*

**H5.** *Performance in health systems is indirectly dependent on the level of final consumption expenditure of households of a state;*

**H6.** *Performance in health systems is directly dependent on the dynamics of a country's gross domestic product.*

To demonstrate the working hypotheses, the multiple linear regression method was used, selecting as dependent variables *CHE*—current health expenditure by financing schemes, *GSCCHFS*—government schemes and compulsory contributory health care financing schemes, respectively *OOPS*—household out-of-pocket payments, specifying that the first dependent variable has a global character and incorporates the other two dependent variables through financing schemes, the modeling aiming to demonstrate the dynamics of performance in the structure. The multiple linear regression method was used because it allows to quantify the correlation relationship based on Pearson approach and to develop the equations models which show usefully information about the trend of indicators in a simple and efficient manner.

## 5. Results and Discussion

Regression equations on states and selected dependent variables were obtained after linear econometric modeling of the database using SPSS, version 25, and are further presented as follows:

$$
\begin{cases}
CHEDenmark = 0.108 * GDPDenmark - 0.145 * PFCDenmark + \\
\quad 0.136 * GGEDenmark + 5.296 * POPDenmark - 33267.895 \\
GSCCHFSDenmark = 0.125 * GDPDenmark - 0.206 * PFCDenmark + \\
\quad 0.129 * GGEDenmark + 3.591 * POPDenmark - 23905.826 \\
OOPSDenmark = -0.014 * GDPDenmark + 0.054 * PFCDenmark + \\
\quad 0.003 * GGEDenmark + 0.827 * POPDenmark - 4570.515
\end{cases}
\tag{1}
$$

$$
\begin{cases}
CHEFrance = -0.04 * GDPFrance + 0.096 * PFCFrance + \\
\quad 0.171 * GGEFrance + 3.403 * POPFrance - 212107.066 \\
GSCCHFSFrance = 0.102 * GDPFrance - 0.199 * PFCFrance + \\
\quad 0.144 * GGEFrance + 6.56 * POPFrance - 389374.392 \\
OOPSFrance = -0.053 * GDPFrance + 0.105 * PFCFrance + \\
\quad 0.012 * GGEFrance + 1.063 * POPFrance - 70408.855
\end{cases}
\tag{2}
$$

$$\begin{cases} CHESpain = 0.142 * GDPSpain - 0.192 * PFCSpain + \\ \quad 0.087 * GGESpain + 4.456 * POPSpain - 177505.772 \\ GSCCHFSSpain = 0.123 * GDPSpain - 0.186 * PFCSpain + \\ \quad 0.096 * GGESpain + 2.026 * POPSpain - 83756.296 \\ OOPSSpain = -0.01 * GDPSpain + 0.034 * PFCSpain + \\ \quad 0.001 * GGESpain + 1.574 * POPSpain - 59771.043 \end{cases} \quad (3)$$

$$\begin{cases} CHEHungary = -0.097 * GDPHungary + 0.212 * PFCHungary + \\ \quad 0.108 * GGEHungary - 2.667 * POPHungary + 27302.453 \\ GSCCHFSHungary = -0.047 * GDPHungary + 0.103 * PFCHungary + \\ \quad 0.083 * GGEHungary - 0.753 * POPHungary + 7777.71 \\ OOPSHungary = -0.038 * GDPHungary + 0.086 * PFCHungary + \\ \quad 0.017 * GGEHungary - 1.783 * POPHungary + 18322.365 \end{cases} \quad (4)$$

$$\begin{cases} CHEBulgaria = 0.083 * GDPBulgaria - 0.083 * PFCBulgaria + \\ \quad 0.066 * GGEBulgaria - 1.019 * POPBulgaria + 8348.905 \\ GSCCHFSBulgaria = 0.06 * GDPBulgaria - 0.096 * PFCBulgaria + \\ \quad 0.079 * GGEBulgaria - 0.696 * POPBulgaria + 5762.648 \\ OOPSBulgaria = 0.022 * GDPBulgaria + 0.013 * PFCBulgaria - \\ \quad 0.012 * GGEBulgaria - 0.318 * POPBulgaria + 2536.455 \end{cases} \quad (5)$$

$$\begin{cases} CHERomania = 0.003 * GDPRomania + 0.029 * PFCRomania + \\ \quad 0.08 * GGERomania - 0.443 * POPRomania + 9366.713 \\ GSCCHFSRomania = 0.009 * GDPRomania + 0.019 * PFCRomania + \\ \quad 0.062 * GGERomania - 0.093 * POPRomania + 1594.978 \\ OOPSRomania = -0.005 * GDPRomania + 0.008 * PFCRomania + \\ \quad 0.019 * GGERomania - 0.321 * POPRomania + 7157.556 \end{cases} \quad (6)$$

From the regression equations, it can be observed that the first three countries present in the HCI ranking—France, Denmark and Spain display negative values for the residual coefficients, both on the global component of expenses CHE—current health expenditure by financing schemes, as well as on components as GSCCHFS—government schemes and compulsory contributory health care financing schemes, OOPS—household out-of-pocket payments. On the other hand, for the states in the tail of the ranking, the values of the residual coefficients are positive. This aspect demonstrates that the regression equations are better determined in terms of performance for the first ranked states and less determined for the last three ranked states.

Regarding the coefficients of the GDP regressor, it can be observed that both in the general expenditure model and in that of the GSCCHFS component—government schemes and compulsory contributory health care financing schemes, there is a direct proportional relationship between the increase in GDP and the increase in financial allocations for health. In case of the OOPS—household out-of-pocket payments component, the correlation trend of the regressor with the dependent variable becomes negative with only one exception in the case of Bulgaria, which has the most ineffective system of financial allocations among the analyzed states. Both in case of the general model and of the specific models, it is observed that the first three states present a higher correlation of GDP with the dependent variables, respectively, the financing schemes tend to take more into account the economic evolution in conditions of uncertainty, while the last three states tend to harden health policies to funding schemes insensitive to uncertainty factors as a result of the inadequate level of funding they provide to the system. The working hypothesis H6 is thus confirmed.

Regarding the final indicator consumption expenditure of households, it seems to be in direct correlation with the private component of financing and tends to correlate inversely proportionally with the governmental component. Hungary and Romania show different trends on the GSCCHFS component—government schemes and compulsory contributory health care financing schemes, which take the capital shocks expressed by

changing population consumption and accumulate them in the government accumulation trend in a direct proportional relationship, confirming the working hypothesis H5.

Regarding the GGE—general government expenditure component, they show a more significant direct correlation with the general model of allocations and with the government model, the values of the correlation coefficients decreasing significantly in case of the model of private allocations. The highest correlation coefficients are found for the first three ranking states, with the intensity of the correlation decreasing for the last three ranked states, which confirms the working hypothesis H4.

The conceptualized models demonstrate a level of statistical significance above 95% with a minimal standard error of the estimator, a number of four degrees of freedom associated with the regression and a significantly higher F-function value for the states ranked in the top three places of the ranking compared to the other three states analyzed.

The errors significance's coefficient allows the validation of the alternative hypothesis and the rejection of the null hypothesis, which further confirms the validity of the performance model of allocations in health systems (see Table 2 below).

**Table 2.** Summary of allocation performance models in health systems.

| Model [a] | Dependent Variable: | R | R Square | Adjusted R Square | Std. Error of the Estimate | R Square Change | Change Statistics F Change | df1 | df2 | Sig. F Change | Durbin-Watson |
|---|---|---|---|---|---|---|---|---|---|---|---|
| | CHE | 0.992 | 0.984 | 0.981 | 213.23105 | 0.984 | 266.313 | 4 | 17 | 0.000 | 1.494 |
| Bulgaria | GSCCHFS | 0.985 | 0.971 | 0.964 | 176.64837 | 0.971 | 142.687 | 4 | 17 | 0.000 | 0.938 |
| | OOPS | 0.983 | 0.967 | 0.959 | 122.34340 | 0.967 | 123.972 | 4 | 17 | 0.000 | 1.198 |
| | CHE | 0.998 | 0.996 | 0.995 | 605.78836 | 0.996 | 980.634 | 4 | 17 | 0.000 | 1.763 |
| Denmark | GSCCHFS | 0.997 | 0.995 | 0.994 | 564.26046 | 0.995 | 821.264 | 4 | 17 | 0.000 | 1.866 |
| | OOPS | 0.998 | 0.996 | 0.995 | 71.28946 | 0.996 | 1026.351 | 4 | 17 | 0.000 | 1.282 |
| | CHE | 1.000 | 1.000 | 0.999 | 1574.75970 | 1.000 | 9320.979 | 4 | 17 | 0.000 | 1.852 |
| France | GSCCHFS | 0.997 | 0.995 | 0.994 | 4367.21661 | 0.995 | 846.646 | 4 | 17 | 0.000 | 1.860 |
| | OOPS | 0.989 | 0.979 | 0.973 | 1367.99689 | 0.979 | 193.579 | 4 | 17 | 0.000 | 1.068 |
| | CHE | 0.993 | 0.986 | 0.983 | 302.98599 | 0.986 | 307.026 | 4 | 17 | 0.000 | 1.802 |
| Hungary | GSCCHFS | 0.986 | 0.972 | 0.965 | 297.03536 | 0.972 | 146.847 | 4 | 17 | 0.000 | 1.328 |
| | OOPS | 0.989 | 0.979 | 0.974 | 102.92694 | 0.979 | 195.147 | 4 | 17 | 0.000 | 1.412 |
| | CHE | 0.987 | 0.975 | 0.969 | 736.87522 | 0.975 | 163.323 | 4 | 17 | 0.000 | 0.929 |
| Romania | GSCCHFS | 0.983 | 0.966 | 0.958 | 680.65101 | 0.966 | 120.897 | 4 | 17 | 0.000 | 1.044 |
| | OOPS | 0.993 | 0.987 | 0.984 | 105.17957 | 0.987 | 317.622 | 4 | 17 | 0.000 | 1.821 |
| | CHE | 0.999 | 0.998 | 0.997 | 1726.56440 | 0.998 | 1851.482 | 4 | 17 | 0.000 | 2.084 |
| Spain | GSCCHFS | 0.999 | 0.998 | 0.998 | 1164.07642 | 0.998 | 2200.120 | 4 | 17 | 0.000 | 1.639 |
| | OOPS | 0.993 | 0.987 | 0.983 | 817.94204 | 0.987 | 312.926 | 4 | 17 | 0.000 | 1.548 |

[a] Predictors: (Constant), POP, GGE, GDP, PFC.

The autocorrelation test based on the following parameters was applied to highlight the homogeneity of the data used in the modeling: maximum number of lags in autocorrelation or partial autocorrelation plots (MXAUTO) = 16; maximum number of lags per cross-correlation plots (MXCROSS) = 7; confidence interval percentage value (CIN) = 95%; method of calculating std. errors for autocorrelations (ACFSE) = independence model. Series length = 22; number of missing values: user-missing = 0; system-missing = 0; number of valid values = 22 (Table 3).

**Table 3.** Autocorrelations.

| Lag | Autocorrelations [a] | Bulgaria | | Denmark | | France | | Hungary | | Romania | | Spain | |
|---|---|---|---|---|---|---|---|---|---|---|---|---|---|
| | | A * | SE ** | A * | SE ** | A * | SE ** | A * | SE ** | A * | SE ** | A * | SE ** |
| 1 | | −0.232 | 0.218 | 0.179 | 0.218 | 0.078 | 0.218 | 0.059 | 0.218 | 0.121 | 0.218 | 0.301 | 0.218 |
| 2 | | 0.010 | 0.230 | −0.064 | 0.225 | −0.128 | 0.220 | −0.036 | 0.219 | −0.122 | 0.221 | 0.077 | 0.237 |
| 3 | | 0.191 | 0.230 | 0.202 | 0.226 | 0.222 | 0.223 | 0.398 | 0.219 | −0.088 | 0.225 | 0.292 | 0.238 |
| 4 | | 0.004 | 0.237 | 0.186 | 0.234 | 0.133 | 0.233 | 0.006 | 0.251 | −0.240 | 0.226 | 0.094 | 0.255 |
| 5 | | −0.466 | 0.237 | −0.002 | 0.241 | 0.069 | 0.237 | −0.115 | 0.251 | −0.019 | 0.238 | −0.037 | 0.257 |
| 6 | | 0.216 | 0.277 | −0.334 | 0.241 | −0.216 | 0.238 | 0.027 | 0.254 | 0.061 | 0.238 | −0.253 | 0.257 |
| 7 | | −0.258 | 0.285 | −0.179 | 0.262 | −0.122 | 0.247 | −0.146 | 0.254 | −0.243 | 0.239 | −0.292 | 0.268 |
| 8 | CHE | −0.201 | 0.296 | −0.279 | 0.268 | −0.251 | 0.250 | −0.211 | 0.258 | −0.298 | 0.250 | −0.358 | 0.283 |
| 9 | | 0.137 | 0.303 | −0.194 | 0.282 | −0.138 | 0.262 | −0.143 | 0.266 | −0.084 | 0.267 | −0.273 | 0.304 |
| 10 | | 0.015 | 0.305 | 0.011 | 0.288 | 0.068 | 0.265 | −0.031 | 0.270 | 0.209 | 0.268 | −0.035 | 0.315 |
| 11 | | −0.105 | 0.306 | −0.207 | 0.288 | −0.210 | 0.266 | −0.216 | 0.270 | 0.082 | 0.276 | −0.151 | 0.316 |
| 12 | | 0.109 | 0.307 | −0.228 | 0.295 | −0.267 | 0.274 | −0.199 | 0.278 | 0.164 | 0.277 | −0.190 | 0.319 |
| 13 | | 0.091 | 0.309 | 0.086 | 0.303 | 0.016 | 0.286 | −0.005 | 0.285 | 0.061 | 0.281 | 0.059 | 0.324 |
| 14 | | −0.012 | 0.310 | 0.176 | 0.304 | 0.147 | 0.286 | 0.003 | 0.285 | −0.031 | 0.282 | 0.128 | 0.325 |
| 15 | | 0.006 | 0.310 | 0.010 | 0.309 | −0.002 | 0.289 | 0.053 | 0.285 | 0.031 | 0.282 | 0.047 | 0.327 |
| 16 | | −0.011 | 0.310 | −0.038 | 0.309 | −0.044 | 0.289 | −0.015 | 0.285 | −0.058 | 0.282 | −0.020 | 0.328 |
| 1 | | −0.202 | 0.218 | 0.198 | 0.218 | −0.033 | 0.218 | 0.013 | 0.218 | 0.077 | 0.218 | 0.360 | 0.218 |
| 2 | | −0.066 | 0.227 | −0.052 | 0.227 | −0.252 | 0.218 | 0.027 | 0.218 | −0.107 | 0.219 | 0.200 | 0.245 |
| 3 | | 0.202 | 0.228 | 0.183 | 0.227 | 0.115 | 0.232 | 0.354 | 0.218 | −0.119 | 0.222 | 0.258 | 0.253 |
| 4 | | −0.041 | 0.236 | 0.180 | 0.234 | 0.104 | 0.235 | −0.039 | 0.244 | −0.279 | 0.225 | 0.041 | 0.265 |
| 5 | | −0.407 | 0.237 | −0.012 | 0.241 | 0.109 | 0.237 | −0.122 | 0.244 | 0.057 | 0.241 | −0.072 | 0.265 |
| 6 | | 0.245 | 0.268 | −0.344 | 0.241 | −0.144 | 0.239 | 0.013 | 0.247 | 0.073 | 0.242 | −0.270 | 0.266 |
| 7 | | −0.091 | 0.278 | −0.192 | 0.263 | −0.062 | 0.243 | −0.148 | 0.247 | −0.216 | 0.243 | −0.330 | 0.279 |
| 8 | GSCCHFS | −0.270 | 0.280 | −0.272 | 0.270 | −0.210 | 0.244 | −0.235 | 0.252 | −0.263 | 0.252 | −0.406 | 0.297 |
| 9 | | 0.099 | 0.292 | −0.184 | 0.282 | −0.012 | 0.252 | −0.107 | 0.262 | −0.126 | 0.264 | −0.261 | 0.322 |
| 10 | | 0.054 | 0.294 | −0.012 | 0.288 | 0.109 | 0.252 | −0.021 | 0.264 | 0.202 | 0.267 | −0.110 | 0.332 |
| 11 | | −0.093 | 0.294 | −0.213 | 0.288 | −0.166 | 0.255 | −0.238 | 0.264 | 0.102 | 0.274 | −0.128 | 0.334 |
| 12 | | 0.071 | 0.295 | −0.205 | 0.296 | −0.250 | 0.260 | −0.087 | 0.274 | 0.166 | 0.276 | −0.092 | 0.336 |
| 13 | | 0.050 | 0.296 | 0.097 | 0.302 | 0.012 | 0.271 | −0.010 | 0.275 | 0.055 | 0.281 | 0.068 | 0.337 |
| 14 | | −0.041 | 0.297 | 0.168 | 0.304 | 0.125 | 0.271 | −0.004 | 0.275 | −0.064 | 0.281 | 0.097 | 0.338 |
| 15 | | 0.026 | 0.297 | 0.010 | 0.308 | −0.018 | 0.274 | 0.091 | 0.275 | 0.038 | 0.282 | 0.041 | 0.339 |
| 16 | | −0.022 | 0.297 | −0.030 | 0.308 | −0.031 | 0.274 | −0.022 | 0.277 | −0.071 | 0.282 | 0.007 | 0.339 |
| 1 | | −0.029 | 0.218 | 0.119 | 0.218 | 0.102 | 0.218 | −0.011 | 0.218 | −0.072 | 0.218 | 0.041 | 0.218 |
| 2 | | −0.058 | 0.218 | −0.131 | 0.221 | −0.006 | 0.220 | −0.144 | 0.218 | 0.001 | 0.219 | −0.273 | 0.219 |
| 3 | | 0.085 | 0.219 | 0.264 | 0.225 | 0.195 | 0.221 | 0.382 | 0.223 | 0.072 | 0.219 | 0.218 | 0.234 |
| 4 | | 0.098 | 0.221 | 0.194 | 0.239 | 0.005 | 0.229 | 0.014 | 0.252 | −0.027 | 0.220 | 0.191 | 0.244 |
| 5 | | −0.146 | 0.223 | 0.025 | 0.247 | 0.097 | 0.229 | −0.073 | 0.252 | −0.201 | 0.221 | 0.063 | 0.251 |
| 6 | | −0.021 | 0.227 | −0.241 | 0.247 | −0.360 | 0.230 | 0.056 | 0.253 | −0.216 | 0.229 | −0.151 | 0.251 |
| 7 | | −0.394 | 0.227 | −0.060 | 0.258 | −0.142 | 0.256 | −0.103 | 0.254 | 0.026 | 0.239 | −0.143 | 0.256 |
| 8 | OOPS | −0.177 | 0.258 | −0.266 | 0.258 | −0.104 | 0.260 | −0.150 | 0.256 | −0.458 | 0.239 | −0.170 | 0.260 |
| 9 | | 0.053 | 0.264 | −0.243 | 0.271 | −0.365 | 0.262 | −0.105 | 0.260 | 0.122 | 0.277 | −0.197 | 0.265 |
| 10 | | 0.003 | 0.264 | 0.087 | 0.281 | 0.054 | 0.285 | −0.055 | 0.262 | 0.117 | 0.280 | 0.132 | 0.272 |
| 11 | | −0.088 | 0.264 | −0.146 | 0.283 | −0.135 | 0.285 | −0.066 | 0.262 | −0.036 | 0.282 | −0.072 | 0.275 |
| 12 | | −0.005 | 0.265 | −0.293 | 0.286 | −0.196 | 0.288 | −0.329 | 0.263 | 0.079 | 0.283 | −0.394 | 0.276 |
| 13 | | 0.123 | 0.265 | −0.002 | 0.300 | 0.052 | 0.295 | −0.028 | 0.282 | 0.063 | 0.284 | 0.045 | 0.301 |
| 14 | | 0.036 | 0.268 | 0.183 | 0.300 | 0.103 | 0.295 | 0.043 | 0.282 | 0.156 | 0.284 | 0.149 | 0.302 |
| 15 | | −0.005 | 0.268 | 0.005 | 0.305 | 0.184 | 0.297 | −0.018 | 0.282 | −0.080 | 0.288 | 0.016 | 0.305 |
| 16 | | 0.014 | 0.268 | −0.100 | 0.305 | −0.043 | 0.302 | 0.002 | 0.283 | 0.051 | 0.289 | −0.068 | 0.305 |
| 1 | | 0.112 | 0.218 | 0.031 | 0.218 | 0.027 | 0.218 | 0.034 | 0.218 | 0.138 | 0.218 | 0.131 | 0.218 |
| 2 | | −0.177 | 0.221 | −0.261 | 0.218 | −0.251 | 0.218 | −0.303 | 0.218 | −0.296 | 0.222 | −0.074 | 0.222 |
| 3 | | 0.186 | 0.228 | 0.244 | 0.233 | 0.236 | 0.232 | 0.340 | 0.238 | 0.054 | 0.240 | 0.264 | 0.223 |
| 4 | | −0.077 | 0.235 | 0.124 | 0.245 | 0.131 | 0.243 | 0.065 | 0.260 | −0.187 | 0.241 | 0.108 | 0.237 |
| 5 | | −0.028 | 0.236 | −0.011 | 0.248 | 0.104 | 0.246 | −0.088 | 0.261 | −0.062 | 0.248 | 0.062 | 0.240 |
| 6 | | −0.165 | 0.236 | −0.121 | 0.248 | −0.188 | 0.248 | 0.026 | 0.262 | 0.060 | 0.249 | −0.234 | 0.241 |
| 7 | | −0.254 | 0.241 | −0.117 | 0.250 | −0.109 | 0.255 | −0.143 | 0.262 | −0.152 | 0.249 | −0.175 | 0.251 |
| 8 | GDP | −0.240 | 0.254 | −0.178 | 0.253 | −0.142 | 0.257 | −0.236 | 0.266 | −0.188 | 0.254 | −0.160 | 0.257 |
| 9 | | −0.155 | 0.264 | −0.190 | 0.259 | −0.235 | 0.261 | −0.196 | 0.276 | −0.135 | 0.260 | −0.302 | 0.262 |
| 10 | | 0.152 | 0.269 | 0.142 | 0.266 | 0.155 | 0.271 | 0.087 | 0.282 | 0.131 | 0.263 | 0.082 | 0.278 |
| 11 | | 0.062 | 0.273 | −0.125 | 0.269 | −0.085 | 0.275 | 0.067 | 0.283 | 0.140 | 0.267 | −0.065 | 0.279 |
| 12 | | −0.105 | 0.273 | −0.328 | 0.272 | −0.369 | 0.276 | −0.250 | 0.284 | −0.073 | 0.270 | −0.331 | 0.280 |
| 13 | | 0.068 | 0.275 | 0.046 | 0.290 | −0.011 | 0.299 | −0.049 | 0.294 | 0.031 | 0.271 | −0.025 | 0.298 |
| 14 | | 0.138 | 0.276 | 0.172 | 0.290 | 0.173 | 0.299 | 0.128 | 0.295 | 0.071 | 0.271 | 0.133 | 0.298 |
| 15 | | 0.022 | 0.279 | 0.016 | 0.295 | 0.027 | 0.304 | 0.027 | 0.297 | −0.011 | 0.272 | 0.033 | 0.301 |
| 16 | | −0.031 | 0.279 | −0.057 | 0.295 | −0.083 | 0.304 | −0.022 | 0.298 | −0.018 | 0.272 | −0.061 | 0.301 |

**Table 3.** *Cont.*

| Lag | Autocorrelations [a] | Bulgaria | | Denmark | | France | | Hungary | | Romania | | Spain | |
|---|---|---|---|---|---|---|---|---|---|---|---|---|---|
| | | A * | SE ** | A * | SE ** | A * | SE ** | A * | SE ** | A * | SE ** | A * | SE ** |
| 1 | | 0.182 | 0.218 | 0.082 | 0.218 | 0.023 | 0.218 | 0.103 | 0.218 | 0.159 | 0.218 | 0.062 | 0.218 |
| 2 | | −0.219 | 0.225 | −0.229 | 0.220 | −0.207 | 0.218 | −0.264 | 0.221 | −0.286 | 0.224 | −0.135 | 0.219 |
| 3 | | 0.162 | 0.235 | 0.199 | 0.231 | 0.239 | 0.227 | 0.292 | 0.235 | 0.076 | 0.240 | 0.249 | 0.223 |
| 4 | | 0.120 | 0.241 | 0.149 | 0.239 | 0.122 | 0.239 | 0.094 | 0.252 | −0.134 | 0.242 | 0.139 | 0.236 |
| 5 | | −0.092 | 0.243 | 0.063 | 0.243 | 0.143 | 0.242 | −0.004 | 0.253 | −0.122 | 0.245 | 0.059 | 0.240 |
| 6 | | −0.133 | 0.245 | −0.154 | 0.244 | −0.220 | 0.246 | 0.010 | 0.253 | 0.005 | 0.248 | −0.217 | 0.240 |
| 7 | | −0.161 | 0.248 | −0.130 | 0.248 | −0.138 | 0.255 | −0.223 | 0.253 | −0.126 | 0.248 | −0.112 | 0.250 |
| 8 | | −0.350 | 0.253 | −0.203 | 0.252 | −0.126 | 0.259 | −0.248 | 0.263 | −0.205 | 0.251 | −0.147 | 0.252 |
| 9 | PFC | −0.192 | 0.275 | −0.176 | 0.259 | −0.241 | 0.262 | −0.146 | 0.274 | −0.154 | 0.259 | −0.281 | 0.256 |
| 10 | | 0.103 | 0.282 | 0.127 | 0.265 | 0.138 | 0.272 | 0.037 | 0.277 | 0.152 | 0.263 | 0.092 | 0.270 |
| 11 | | 0.018 | 0.283 | −0.145 | 0.268 | −0.107 | 0.275 | 0.005 | 0.277 | 0.181 | 0.267 | −0.043 | 0.272 |
| 12 | | −0.186 | 0.284 | −0.329 | 0.272 | −0.367 | 0.277 | −0.265 | 0.277 | −0.101 | 0.273 | −0.354 | 0.272 |
| 13 | | 0.048 | 0.289 | 0.021 | 0.290 | 0.006 | 0.300 | −0.061 | 0.289 | 0.012 | 0.275 | −0.026 | 0.293 |
| 14 | | 0.170 | 0.290 | 0.178 | 0.290 | 0.166 | 0.300 | 0.139 | 0.290 | 0.082 | 0.275 | 0.140 | 0.293 |
| 15 | | 0.019 | 0.294 | 0.016 | 0.295 | 0.018 | 0.304 | 0.026 | 0.293 | −0.030 | 0.276 | 0.020 | 0.296 |
| 16 | | −0.018 | 0.294 | −0.071 | 0.295 | −0.081 | 0.304 | −0.013 | 0.293 | −0.020 | 0.276 | −0.074 | 0.297 |
| 1 | | 0.010 | 0.218 | 0.145 | 0.218 | 0.127 | 0.218 | 0.166 | 0.218 | 0.178 | 0.218 | 0.368 | 0.218 |
| 2 | | −0.294 | 0.218 | −0.007 | 0.223 | −0.144 | 0.222 | −0.171 | 0.224 | −0.089 | 0.225 | 0.208 | 0.246 |
| 3 | | −0.162 | 0.236 | 0.117 | 0.223 | 0.206 | 0.226 | 0.324 | 0.230 | 0.017 | 0.227 | 0.197 | 0.254 |
| 4 | | −0.026 | 0.242 | 0.205 | 0.226 | 0.163 | 0.235 | 0.040 | 0.251 | −0.199 | 0.227 | 0.125 | 0.262 |
| 5 | | −0.021 | 0.242 | −0.089 | 0.234 | 0.034 | 0.240 | −0.146 | 0.251 | −0.262 | 0.235 | −0.118 | 0.264 |
| 6 | | 0.295 | 0.242 | −0.150 | 0.236 | −0.171 | 0.240 | 0.126 | 0.255 | −0.020 | 0.248 | −0.244 | 0.267 |
| 7 | | −0.067 | 0.258 | −0.164 | 0.240 | −0.175 | 0.246 | −0.100 | 0.258 | −0.103 | 0.249 | −0.359 | 0.277 |
| 8 | GGE | −0.440 | 0.259 | −0.287 | 0.246 | −0.281 | 0.252 | −0.410 | 0.260 | −0.275 | 0.251 | −0.437 | 0.299 |
| 9 | | 0.001 | 0.293 | −0.072 | 0.261 | −0.090 | 0.266 | −0.151 | 0.289 | −0.084 | 0.265 | −0.172 | 0.328 |
| 10 | | 0.051 | 0.293 | 0.024 | 0.262 | 0.040 | 0.268 | −0.022 | 0.293 | −0.014 | 0.266 | −0.147 | 0.332 |
| 11 | | 0.106 | 0.293 | −0.196 | 0.262 | −0.209 | 0.268 | −0.104 | 0.293 | 0.082 | 0.266 | −0.171 | 0.335 |
| 12 | | −0.020 | 0.295 | −0.234 | 0.269 | −0.280 | 0.276 | −0.114 | 0.295 | 0.267 | 0.267 | −0.080 | 0.339 |
| 13 | | 0.073 | 0.295 | 0.010 | 0.279 | 0.034 | 0.289 | −0.065 | 0.297 | 0.119 | 0.280 | 0.070 | 0.340 |
| 14 | | −0.007 | 0.296 | 0.121 | 0.279 | 0.146 | 0.289 | 0.005 | 0.298 | −0.020 | 0.282 | 0.104 | 0.341 |
| 15 | | 0.043 | 0.296 | −0.018 | 0.281 | −0.006 | 0.293 | 0.054 | 0.298 | −0.010 | 0.282 | 0.027 | 0.342 |
| 16 | | −0.005 | 0.296 | −0.032 | 0.281 | −0.036 | 0.293 | 0.048 | 0.298 | −0.036 | 0.282 | 0.038 | 0.342 |
| 1 | | −0.039 | 0.218 | 0.699 | 0.218 | 0.085 | 0.218 | 0.076 | 0.218 | −0.046 | 0.218 | 0.610 | 0.218 |
| 2 | | 0.039 | 0.219 | 0.304 | 0.309 | −0.031 | 0.220 | −0.137 | 0.219 | −0.050 | 0.219 | 0.424 | 0.288 |
| 3 | | 0.021 | 0.219 | −0.080 | 0.323 | −0.009 | 0.220 | −0.169 | 0.224 | −0.033 | 0.219 | 0.258 | 0.316 |
| 4 | | 0.030 | 0.219 | −0.287 | 0.324 | −0.100 | 0.220 | −0.015 | 0.230 | −0.085 | 0.219 | 0.100 | 0.326 |
| 5 | | 0.011 | 0.219 | −0.217 | 0.336 | −0.177 | 0.222 | −0.099 | 0.230 | −0.033 | 0.221 | −0.109 | 0.328 |
| 6 | | −0.007 | 0.219 | −0.036 | 0.343 | −0.100 | 0.229 | −0.164 | 0.232 | 0.280 | 0.221 | −0.311 | 0.330 |
| 7 | | −0.033 | 0.219 | 0.167 | 0.343 | 0.240 | 0.231 | −0.263 | 0.237 | −0.139 | 0.238 | −0.463 | 0.343 |
| 8 | POP | −0.057 | 0.219 | 0.177 | 0.347 | 0.016 | 0.243 | −0.233 | 0.251 | −0.079 | 0.241 | −0.443 | 0.372 |
| 9 | | −0.033 | 0.220 | 0.039 | 0.351 | −0.033 | 0.243 | 0.348 | 0.261 | −0.098 | 0.243 | −0.366 | 0.396 |
| 10 | | −0.074 | 0.220 | −0.170 | 0.351 | −0.067 | 0.243 | 0.132 | 0.282 | −0.101 | 0.244 | −0.352 | 0.412 |
| 11 | | −0.079 | 0.222 | −0.327 | 0.355 | −0.113 | 0.244 | 0.035 | 0.285 | −0.105 | 0.246 | −0.277 | 0.426 |
| 12 | | −0.095 | 0.223 | −0.393 | 0.369 | −0.100 | 0.246 | 0.020 | 0.285 | −0.096 | 0.249 | −0.085 | 0.434 |
| 13 | | −0.074 | 0.225 | −0.305 | 0.389 | −0.108 | 0.248 | 0.099 | 0.285 | 0.144 | 0.250 | 0.068 | 0.435 |
| 14 | | −0.056 | 0.226 | −0.184 | 0.400 | −0.094 | 0.250 | −0.033 | 0.287 | −0.035 | 0.254 | 0.070 | 0.436 |
| 15 | | −0.043 | 0.227 | −0.088 | 0.404 | −0.040 | 0.252 | −0.012 | 0.287 | −0.048 | 0.254 | 0.081 | 0.436 |
| 16 | | −0.049 | 0.227 | 0.009 | 0.405 | 0.019 | 0.252 | −0.117 | 0.287 | −0.049 | 0.255 | 0.116 | 0.437 |

[a] The underlying process assumed is MA with the order equal to the lag number minus one. The Bartlett approximation is used. * A—autocorrelation; ** SE—std. error.

According to the autocorrelation table (Table 3), the variables are independent of each other and the autocorrelation phenomenon does not exist (autocorrelation coefficients have values between −0.5 and 0.5, with two exceptions for the POP indicator).

Analysis of the distribution through the P-P plot graph in the case of the general model shows that the uncertainties had different effects on the performance models of the allocations in the health systems (Figure 7).

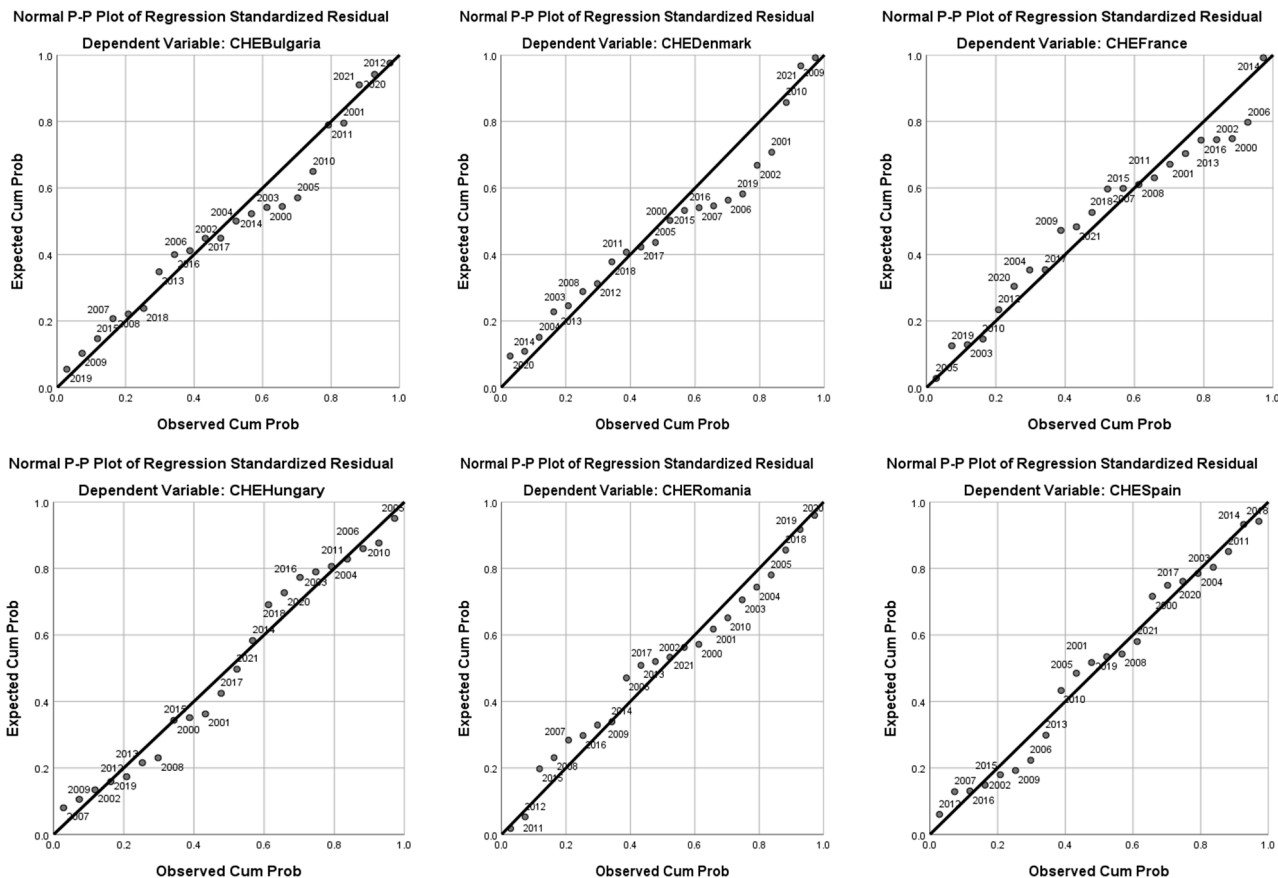

**Figure 7.** The P-P plot diagram of the general regression model for the analyzed states. Source: the authors using IBM SPSS vs. 25.

In case of Bulgaria, the 2000–2005 period (the pre-accession period to the EU) generated the highest distribution of the model trend, followed by the 2008–2011 economic crisis. The impact of the COVID-19 pandemic did not generate errors in the distribution of the observed variable compared to the right of the trend.

In Denmark's case, the main phenomena that changed the observed distribution of the dependent variable to the right of the trend were the accession to the Schengen area (2001), the period of the economic crisis (2008–2011) and the pandemic period, of 2019–2021.

In France the series of observations showed significant deviations from trend curve, as during the 2000–2002 period, due to social and political movements, in the period of the global economic crisis (2008–2011), during the 2016 social crisis and within the pandemic crisis.

Compared to the states presented previously, Hungary presents a greater homogeneity of the data compared to the trend line assimilated to a more general opposability of the model. Here there are some influences for the periods of 2000–2001 and the pandemic of 2019–2021, but the level of errors is lower. This signifies a rigidity of the allocations with the minimum stability threshold and an exclusion of the variation under uncertainty.

Romania shows a similar situation to Hungary, although there are uncertainties' influences from the pre-accession, the global economic crisis and the pandemic crisis' periods.

Spain's model, similar to the one of France, shows uncertainty influences for the global economic crisis and the pandemic periods.

Through the ANOVA test, the validity of the econometric models was confirmed, with the sum of the squares of the regressions being strictly higher than the residual component, both in case of the general model and of the two component models, as it can be seen from Table 4 below.

**Table 4.** ANOVA test.

| Model | | Sum of Squares CHE | df CHE | Mean Square CHE | Sum of Squares GSCCHFS | df GSC-CHFS | Mean Square GSCCHFS | Sum of Squares OOPS | df OOPS | Mean Square OOPS |
|---|---|---|---|---|---|---|---|---|---|---|
| Bulgaria | Regression | 48,434,384.9 | 4 | 12,108,596.2 | 17,810,019.5 | 4 | 4,452,504.9 | 7,422,396.7 | 4 | 1,855,599.2 |
| | Residual | 772,947.2 | 17 | 45,467.5 | 530,479.0 | 17 | 31,204.6 | 254,454.4 | 17 | 14,967.9 |
| | Total | 49,207,332.1 | 21 | | 18,340,498.5 | 21 | | 767,6851.1 | 21 | |
| Denmark | Regression | 1,439,491,136.1 | 4 | 359,872,784.0 | 1,045,928,366.1 | 4 | 261,482,091.5 | 20,864,437.0 | 4 | 5,216,109.2 |
| | Residual | 6,238,652.2 | 17 | 366,979.5 | 5,412,627.8 | 17 | 318,389.9 | 86,397.2 | 17 | 5082.2 |
| | Total | 1,445,729,788.3 | 21 | | 1,051,340,993.9 | 21 | | 20,950,834.1 | 21 | |
| France | Regression | 92,459,196,135.4 | 4 | 23,114,799,033.9 | 64,590,896,299.5 | 4 | 16,147,724,074.9 | 1,449,067,050.9 | 4 | 36,226,6762.7 |
| | Residual | 42,157,758.0 | 17 | 2,479,868.1 | 324,233,875.9 | 17 | 19,072,580.9 | 31,814,063.4 | 17 | 1,871,415.5 |
| | Total | 92,501,353,893.4 | 21 | | 64,915,130,175.4 | 21 | | 1,480,881,114.3 | 21 | |
| Hungary | Regression | 112,740,564.6 | 4 | 28,185,141.1 | 51,825,361.0 | 4 | 12,956,340.2 | 8,269,521.8 | 4 | 2,067,380.5 |
| | Residual | 1,560,608.7 | 17 | 91,800.5 | 1,499,910.0 | 17 | 88,230.0 | 1,800,97.2 | 17 | 10,594.0 |
| | Total | 114,301,173.3 | 21 | | 53,325,271.0 | 21 | | 8,449,619.1 | 21 | |
| Romania | Regression | 354,727,327.5 | 4 | 88,681,831.9 | 224,039,576.4 | 4 | 56,009,894.1 | 14,055,078.2 | 4 | 3,513,769.6 |
| | Residual | 9,230,746.6 | 17 | 542,985.1 | 7,875,858.6 | 17 | 463,285.8 | 188,066.6 | 17 | 11,062.7 |
| | Total | 363,958,074.0 | 21 | | 231,915,435.0 | 21 | | 14,243,144.8 | 21 | |
| Spain | Regression | 22,077,249,240.8 | 4 | 5,519,312,310.2 | 11,925,300,378.3 | 4 | 2,981,325,094.6 | 837,425,186.6 | 4 | 209,356,296.7 |
| | Residual | 50,677,418.9 | 17 | 2,981,024.6 | 23,036,256.5 | 17 | 1,355,073.9 | 11,373,496.0 | 17 | 669,029.2 |
| | Total | 22,127,926,659.7 | 21 | | 11,948,336,634.8 | 21 | | 848,798,682.6 | 21 | |

Source: the authors.

The distribution of the quadratic means for both the general model and the component models highlights the performances of the French, Danish and Spanish health systems, while the Hungarian, Romanian and Bulgarian models show a clearly lower efficiency than the first three as can be seen in Figure 8 below.

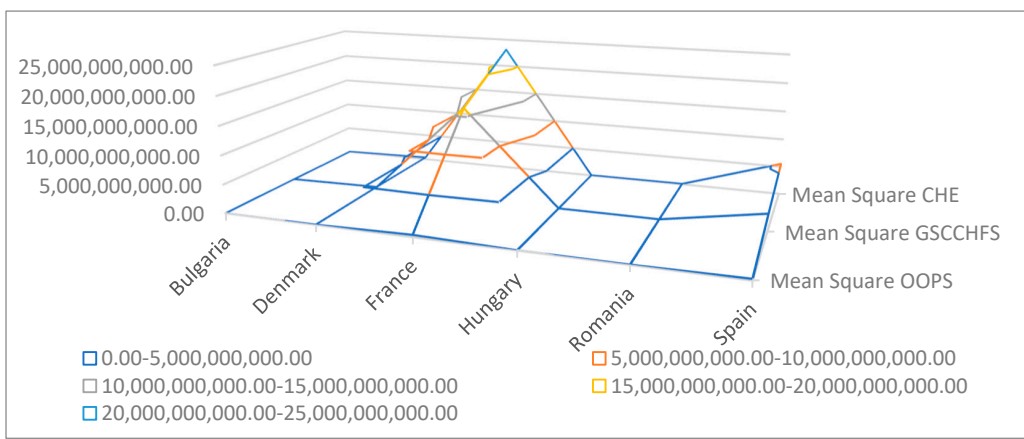

**Figure 8.** The distribution of the square means of the regressions on the analyzed states. Source: the authors.

The design of the residual statistics allowed calculating the public health systems' financing forecast interval for the six states (Appendix A), highlighting the significant disparities in the performance of the six health systems. This confirms the working hypothesis H3, noting that high-performing health systems benefit from a strictly smaller prediction interval amplitude than less-performing systems (Figure 9).

To demonstrate the working hypotheses H1 and H2, the distributions of the forecasting averages were analyzed for the general model and for its components in the Appendix A. It turned out that France, Denmark and Spain benefit in excess of 70% from a distribution of government financial funds whilst direct financing by the population accounts to around 20%. The Spanish model is affected by the influence of the COVID-19 pandemic against the background of demographic characteristics specific to an aging population, an aspect that represented a challenge during the pandemic and changed the predicted values of the

distribution means of the dependent variables of the general model and of the components. For both Bulgaria and Hungary, the allocation performance of government financial funds and the financial burden of medical costs borne directly by patients are in lower parameters than for the efficient financing models of France, Denmark and Spain, which confirms the working hypotheses H1 and H2 (see Figure 10).

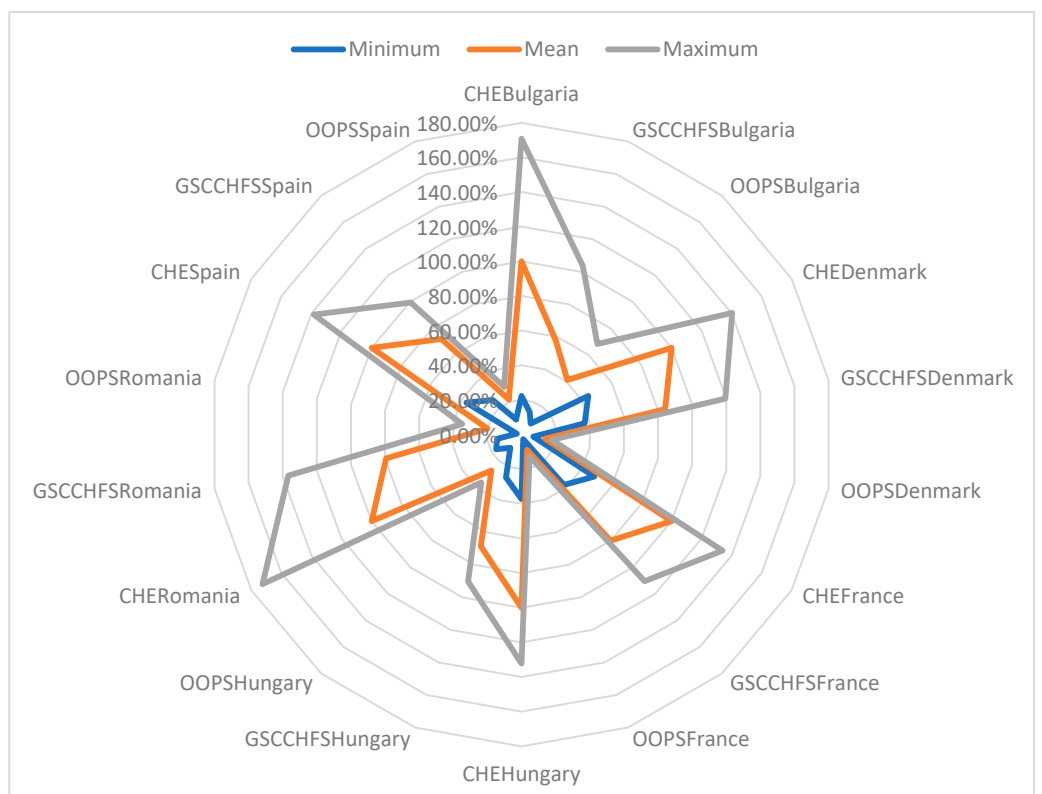

**Figure 9.** Diagram of the projected amplitude of the financing of the health systems in the 6 analyzed states. Source: the authors.

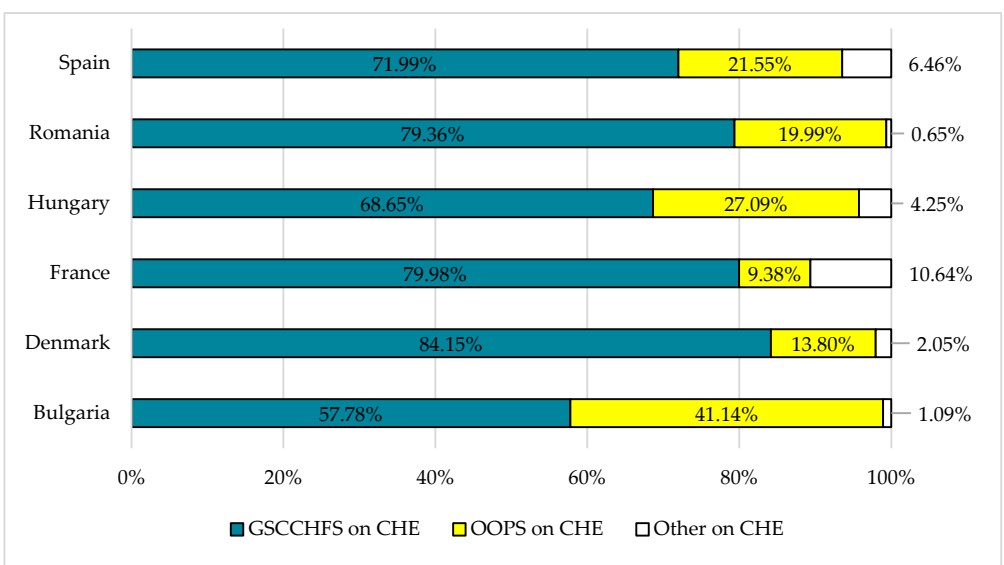

**Figure 10.** The structure of the financing of the health systems in the 6 analyzed states based on the predicted average of the variation of the dependent variables of the regression models by components in relation to the general model. Source: the authors.

Romania is closer to France and Denmark as financing schemes, but this does not make it into a financial performant health system, as it was characterized by a maximum amplitude of the minimum–maximum variation range of the predicted averages as shown in Figure 8.

The COVID-19 pandemic has highlighted the importance of primary health care as an essential component of efficient health systems and basis for universal health coverage (Hanson et al. 2022). Due to low and middle incomes in many countries, like Romania and Bulgaria, not all the conditions for providing quality health services are met due to insufficient public funding, unequal access to services and the fact that patients often have to pay out of pocket. This has led to a vicious circle where underfunded services are unreliable, of poor quality and not accountable to users.

To support the development of people-centered health systems, it is essential to establish appropriate funding mechanisms, including the amount of funding received, methods of funding to service providers and incentives created by payment mechanisms. Both financing and service delivery modalities should be addressed simultaneously. The shocks the economy and the health sector had to face during the last years have exposed the resilience of healthcare systems, highlighting the importance of strategic procurement to improve their efficiency. In this respect, Montás et al. developed a study for six EU countries and found that strategic buyers responded to seven health system shocks, with each country relying on the federal government for funding and response. Buyers often have limited, passive roles in contributing to a resilient health system, the need for strategic procurement in addressing health system challenges being very important (Montás et al. 2022).

The pandemic revealed the real situation of the health systems worldwide. There are significant adverse effects on the global health system, such phenomena having both economic and medical impact, with many negative consequences for the mental health of the population, consequences with significant effects even today (Vasile et al. 2022). Authors like Lupu and Tiganasu analyzed the efficiency of health systems in 31 European countries during the COVID-19 pandemic, using the methodological tools of DEA to calculate efficiency scores for all the health systems analyzed. Six major areas of influence were considered: healthcare, health status, population, economic, cultural/societal and governmental issues. The study showed that Western countries, especially Italy, Belgium, Spain and the UK, experienced high inefficiency in the first phase of the pandemic. However, Western states began to improve their medical systems during the relaxation phase and the second wave, proving their superiority, as proved in the current analysis. The study also found that the influencing factors varied for each stage: population age, population density, government effectiveness and education. The findings can help health policymakers compare good practices and develop national plans to better deal with future health crises (Lupu and Tiganasu 2022).

The global situation characterizing the last years raised questions about global health policy and short-term financing solutions to help health systems cope with shocks. However, research on the relationship between pandemic resilience and long-term health policies is limited. In a recent study, the authors Marginean and Orastean set out to investigate whether the countries of the EU were consistent in terms of financing the national health systems, and how prepared they were to face the shock. The analysis was carried out in 27 countries of the EU, divided according to the level of health expenditure, respectively, high, medium and low. The results showed that countries in the high health spending cluster performed better, indicating that better financing could increase health system performance and resilience to future shocks (Marginean and Orastean 2022).

## 6. Conclusions

The research achieved its objectives by highlighting the concordance between financing and performances of health systems. Extensive research of the specific literature was carried out, pointing out that health sector's financing influences the population's health status. In addition, the econometric modeling demonstrated that the performance in health

is directly dependent on the efficient process of allocating financial funds government (H1). The appropriate allocation of these funds to key health programs and services, such as primary care, emergency services, and disease prevention programs, may improve access to high quality health care services for all citizens. Also, effective management of resources such as medical personnel, equipment and drugs can lead to cost reduction and increased efficiency in the health system. In addition, transparency in the processes of funds' allocating and resources' management can help prevent corruption and ensure fair distribution to all regions and categories of patients. Another hypothesis reveals that the lower the financing performance, the higher the level of direct financing by the population (H2). This phenomenon can be observed in many countries where the performance of state financing is low and the population is forced to rely ever more on direct financing. This can have negative effects on the economy, as the population will have to use their personal savings to cover financial needs, which can lead to lower consumption levels and slower economic growth. Also, the increase in the level of population's direct financing brings about an increase in their indebtedness, especially if there is no adequate financial education. Thus, people may take short-term loans or resort to high-interest loans, leading to increased debt levels and difficulties in paying them off in the future. In addition, the increase in direct financing may also affect the banking sector.

The study allowed the identification of key aspects of health systems' financing and of efficient financing models by referring to the evolution of main macroeconomic indicators, demonstrating that: an efficient financing of the health systems reduces the amplitude margin of the minimum–maximum predicted interval of the financing function (H3); the performance in health systems is directly dependent on the level of public debt of a state (H4); the performance of health systems is indirectly dependent on the level of final consumption of the population of a state (H5); the performance of health systems is directly dependent on the dynamics of the gross domestic product of a state (H6).

Modeling the performance of allocations in the health systems proved that the performance level is higher in the developed states, whereas Romania and Bulgaria displayed the worst performance of the health systems among the six analyzed states. The poor performance of the health systems in Romania and Bulgaria results in a low quality of medical services offered to the population. The lack of adequate funds and infrastructure, as well as the underfunding of the health system, led to a lack of equipment, medicines and qualified medical personnel. In addition, corruption in the healthcare system seriously affects the quality of medical care, which leads to a low level of patient satisfaction. This precarious situation of the health systems in Romania and Bulgaria imposes the need for urgent reforms, to improve the quality of medical services, and ensure access to quality care for all residents. Increased investment in health, including modern infrastructure and equipment, as well as the training and recruitment of well-trained and motivated health personnel, is essential. At the same time, combating corruption in the healthcare system must be a priority, by implementing measures of transparency and financial responsibility towards the citizen. The units considered for this research paper were selected with the purpose of covering two relatively opposite health systems in terms of financing: the countries which allocate relatively high amounts of money towards financing their health expenditures, and the countries with lower shares of resource allocation within their GDP. Moreover, the set of countries may also be grouped into the Bismarck type countries, which finance their health sector mainly through social security contributions to which France, Hungary, Romania and Bulgaria belong, and the Beveridge type countries, which finance their health sector mainly through direct taxes, like Denmark and Spain.

As a result, based on conclusions of this study, we propose the following public policies in order to ensure more sustainable healthcare systems:

1. Increasing efficiency of the financial allocations using new indicators of monitoring.
2. Using predictions regarding demographic trends and chronical diseases structures in establishing the regional financing priorities for healthcare systems.
3. Building healthcare insurance systems in less developed countries able to cover the protection of at risk of poverty population.
4. Applying a dynamic adjustment system of allocations based on health need and the financing disposable sources using reserve funds for crisis situations.
5. Defining practical and efficient measures of preventing illness in order to decrease the pressure on health systems (policies regarding the health culture).
6. Using a double mechanism for financial allocation and collecting in order to ensure the financial equilibrium of the healthcare system.
7. Using ex-ante evaluations of crisis in order to prevent financial deficit in healthcare system.

The limits of the study consist in the relatively small number of analyzed indicators and the limited number of states on which the model was designed, the authors suggesting the expansion of the research in the future in order to complete and fructify the results of the model by transposing it and other perspectives on the performance evolution of the health systems.

**Author Contributions:** Conceptualization, M.S.D., V.M.A., M.L.A., M.R.S. and C.M.B.; Methodology, M.S.D., V.M.A., M.L.A., M.R.S. and C.M.B.; Software, M.S.D., V.M.A., and M.L.A.; Validation, M.L.A., M.R.S. and C.M.B.; Formal Analysis, M.L.A., M.R.S. and C.M.B.; Investigation, M.S.D. and V.M.A.; Resources, M.L.A., M.R.S. and C.M.B.; Data Curation, M.S.D. and V.M.A.; Writing–Original Draft Preparation, M.S.D., V.M.A., M.L.A., M.R.S. and C.M.B.; Writing–Review & Editing, M.S.D., V.M.A., M.L.A., M.R.S. and C.M.B.; Visualization, M.R.S. and C.M.B.; Supervision, M.S.D. and V.M.A.; Project Administration, M.L.A., M.R.S. and C.M.B. All authors have read and agreed to the published version of the manuscript.

**Funding:** This research received no external funding.

**Informed Consent Statement:** Not applicable.

**Data Availability Statement:** The data that support the findings of this study is available from the corresponding author upon request.

**Conflicts of Interest:** The authors declare no conflict of interest.

## Appendix A

| Residuals Statistics | Minimum | Mean | Maximum |
|---|---|---|---|
| CHEBulgaria | 752.1 | 3376.2 | 5768.6 |
| GSCCHFSBulgaria | 463.4 | 1950.7 | 3496.3 |
| OOPSBulgaria | 275.5 | 1388.8 | 2302.8 |
| CHEDenmark | 13,300.3 | 29,885.2 | 41,955.5 |
| GSCCHFSDenmark | 11,068.5 | 25,149.6 | 35,696.7 |
| OOPSDenmark | 2056.9 | 4124.2 | 5305.2 |
| CHEFrance | 129,835.4 | 267,868.2 | 358,867.7 |
| GSCCHFSFrance | 101,892.3 | 214,235.3 | 296,009.4 |
| OOPSFrance | 7716.3 | 25,130.9 | 36158 |
| CHEHungary | 3322.8 | 8932.3 | 11,800.2 |
| GSCCHFSHungary | 2356.6 | 6132.4 | 8070.5 |
| OOPSHungary | 897.6 | 2420.2 | 3231.6 |
| CHERomania | 1436.2 | 8477.1 | 14,640.4 |
| GSCCHFSRomania | 1211.1 | 6727.4 | 11,589 |
| OOPSRomania | 235.4 | 1694.2 | 2948 |
| CHESpain | 39,698.2 | 10,7841.4 | 149,364.8 |
| GSCCHFSSpain | 28,034.2 | 77,638.9 | 107,299.2 |
| OOPSSpain | 10,031 | 23,239.3 | 31,711.6 |

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
