# Peer review of "Modelling Health Financing Performance in Europe in the Context of Macroeconomic Uncertainties"

_economies, doi:10.3390/economies11120299_

Round 1
Reviewer 1 Report
Comments and Suggestions for Authors
Dear authors.
An interesting article which I think will encourage the general public to read it. In my opinion, the article needs some corrections and improvements before proofreading, which I have noticed and suggest below.
1) I suggest that the authors explain at the beginning that this is a comparison of differently developed EU countries, which will most likely be reflected in the organisation of the health system. The same should be explained in the conclusion, where the limitations (of the model) are pointed out.
2) Considering the interesting topic, I lost the thread a few times while reading, mainly because of too extensive paragraphs that are not directly related to the main content of the article. I would suggest that the first section is dedicated to a literature review that is exclusively related to the components used in the model. The general findings about the pandemic described by the authors in the literature review are not directly related to the construction of the model, so I would suggest avoiding this topic. (e.g. p. 13, lines 492-493 and 499-507). If the authors feel that any of the components included in the model are directly or indirectly related to the topic under discussion, they can be included in the discussion.
3) I would suggest the same caution and recommendation for the final section of the article. The authors highlight corruption in the health sector, which is not the subject of the model's analysis (p. 21, lines 764-766 and 758-759). If the authors feel that they are analysing the issue in question in a model, it may be necessary to add a "dummy" variable or similar to the model. Otherwise, I suggest shortening the conclusion and referring to the model result. Basically, the conclusions should be more general, as we know that it is a mathematical modelling tool and an approximation to real life.
After the corrections made, I believe that the article will be interesting, useful and credible for publication in the journal.
Author Response
Dear authors.
An interesting article which I think will encourage the general public to read it. In my opinion, the article needs some corrections and improvements before proofreading, which I have noticed and suggest below.
Dear Reviewer,
First of all, we would like to express our gratitude for the work you dedicated for investigating our research, identifying the points for improvement and suggesting ways for achieving that. We are fully aware that your suggestions and recommendations are very important for improving our research and the way it is presented in this article.
Thank you and we hope we have answered all your suggestions and recommendations and improved our research.
1) I suggest that the authors explain at the beginning that this is a comparison of differently developed EU countries, which will most likely be reflected in the organisation of the health system. The same should be explained in the conclusion, where the limitations (of the model) are pointed out.
Authors: It is a welcome observation. As a result, we made necessary corrections in Abstract (about comparison study) and in Introduction section upside the research objectives, as fallow:
“We started from the premise of considering two opposite group of countries, i.e. France, Denmark, and Spain occupy the first three positions at the level of the EU in terms of economic development, whilst the remaining three countries, respectively Bulgaria, Romania, and Hungary, have common economic characteristics like: geographic position (neighboring states), have joined the EU in the same period, being part of the new Member States (Hungary in 2004, Bulgaria and Romania in 2007), were under communism, and shortly after the fall of communism had adopted similar health systems.”
Moreover, suggested changes and additions have been made in the following paragraphs:
- Introduction: “The six EU Member States for the study were chosen based on the European Health Care Index ranking by country in 2023 produced by the online database (Numbeo, 2023). The Health Care Index is a comparative tool that assesses the overall quality of a health system, including factors such as health professionals, equipment, staff, doctors and costs. The selected countries are differently economic developed countries, with unique features, characterized by certain types of health systems, as seen further in subsection 3.2.”
In the newly created subsection 3.2. [1] The Health Care Index and the Socio-Economic Context of the Selected Countries: “The health care index provides an assessment of healthcare infrastructure, services and resources in different cities or countries, taking an overall snapshot of healthcare in a given location.
According to the data provided by Eurostat (Eurostat, 2023a, 2023b), it can be seen that health expenditures are allocated according to the level of country’s economic development instead of population density” (see Figure 4).
“Denmark has the highest allocation of overall health public expenditure as a percentage of GDP and the lowest number of inhabitants among the analyzed countries. At the same time, indicators such as the inflation rate clearly distinguish developed from less developed countries, with inflation rates nearly half as low as the ones from less developed countries. This can also be seen in the analysis of social welfare as reflected in GDP per capita” (see Figure 5).
“The level of gross public debt in former communist countries is generally lower than in developed countries, as they have less experience with market economies than developed countries and as such the accumulation of public debt reflects this. In terms of budget deficits, with the exception of Denmark, all the other countries analyzed are exposed to budget deficits that generally exceed the 3% threshold of the GDP and reflect the state of uncertainty induced by the multiple crises in the European economies” (see Table 1).
“France's health system incorporates a concept of social health insurance with a significant reliance on social health insurance contributions to finance health services. The health system offers comprehensive coverage to all individuals, encompassing a wide range of benefits. However, individuals are obliged to contribute financially by sharing the costs of the necessary services. The use of complementary private insurance to meet these costs leads to a significant reduction in average expenditure in direct payments (Or et al., 2023). The French health system is of a mixed type, including the Bismarck system (social health insurance), has universal social health insurance, and to ensure financial sustainability, health financing sources have been extended beyond compulsory social contributions to include a wider range of revenue sources, including active financial sources, earmarked investments and taxes and value added taxes. France’s spending on health is among the highest in the European Union at 9.2% of GDP in 2021, above the EU average.
The Danish health system is largely tax-funded, decentralized and organized on three administrative levels: state, regional and local. Planning and regulation take place at both central and local level. The national level is responsible for regulation, supervision, some planning and quality monitoring, while the five regions are responsible for defining and planning health service provision. Municipalities are responsible for health promotion, disease prevention, rehabilitation, home care and long-term care (OECD et al., 2021b). The Danish population is automatically insured under the national health system. Funding is predominantly provided from general tax revenues at state level and, to a lesser extent, from a municipal income tax.
Spain's national health system is based on universal coverage and is mainly funded by social contributions and taxes. The Ministry of Health is responsible for national planning and regulation, health and primary care competences, resource allocation, procurement and delivery are transferred at regional level to the 17 regional health authorities. Financing of the health system is predominantly public, as for private financing, payments come from a combination of out-of-pocket payments and private health insurance (OECD et al., 2021d).
The Bulgarian health system is based on compulsory social health insurance contributions, with voluntary health insurance having a role. The National Health Insurance System of Bulgaria, through its regional health insurance subsidiaries in 28 regions, is the sole purchaser of health services. The Ministry of Health is responsible for the overall governance of the health system, the development of health legislation, the coordination and supervision of various subordinate bodies, and the planning and regulation of health care providers. Even though its health expenditure per capita has increased constantly, Bulgaria still ranks last among EU countries, with direct payments still having a significant weight in health expenditure, (with a share of more than 38%) as can be seen from OECD report - Bulgaria's 2021 Country Profile on Health (OECD et al., 2021a).
The health system in Romania is based on the obligation of social health insurance contributions with a strong involvement of the state. The Ministry of Health is responsible for the general governance of the social health insurance system, while the National Health Insurance House administers and regulates the single national health social insurance fund, through the county directorates of public health and the county health insurance houses. Through the county health insurance companies, medical services are purchased from medical service providers, and the Ministry of Health ensures the payment of national health programs. Even if the social health insurance system in Romania is mandatory, a significant percentage (as stated in the OECD report - Romania's Country Profile of 2021 in terms of health (OECD et al., 2022) of approximately 11% of the population remains uninsured, especially in rural areas. Even if health spending has recently increased in Romania, it still remains one of the EU countries with the lowest health spending, both per capita and as a percentage of GDP.
Hungary has only one health insurance fund that provides medical coverage for the population. The fund is administered by the National Health Insurance Fund Management Institute, which operates under the direct control of the Ministry of Human Resources. The Ministry establishes Development strategies, establishes financing conditions and the package of medical benefits and has a regulatory role. Even though the health system expenditure is provided by public funds, there is still a high level of direct payments for medical services, of about 28%, as shown in the OECD report - Hungary Country Profile 2021 in terms of health (OECD et al., 2021c).”
And in the end of chapter 6. Conclusions: “The units considered for this research paper were selected with the purpose of covering two relatively opposite health systems in terms of financing: the countries which allocate relatively high amounts of money towards financing their health expenditures, and the countries with lower shares of resource allocation within their GDP. Moreover, the set of countries may also be grouped into the Bismarck type countries, which finance their health sector mainly through social security contributions to which France, Hungary, Romania, and Bulgaria belong, and the Beveridge type countries, which finance their health sector mainly through direct taxes, like Denmark and Spain.”
2) Considering the interesting topic, I lost the thread a few times while reading, mainly because of too extensive paragraphs that are not directly related to the main content of the article. I would suggest that the first section is dedicated to a literature review that is exclusively related to the components used in the model. The general findings about the pandemic described by the authors in the literature review are not directly related to the construction of the model, so I would suggest avoiding this topic. (e.g. p. 13, lines 492-493 and 499-507). If the authors feel that any of the components included in the model are directly or indirectly related to the topic under discussion, they can be included in the discussion.
Authors: Suggested modifications and additions have been made.
We took into consideration your remark. As a result, we have modified accordingly the literature overview section. Since there is a connection between this study and the papers discussing the COVID-19 pandemics, we introduced several relevant ideas in the fifth part, in the discussion section.
“The COVID-19 pandemic has highlighted the importance of primary health care as an essential component of efficient health systems and basis for universal health coverage (Hanson et al., 2022). Due to low and middle incomes in many countries, like Romania and Bulgaria, not all the conditions for providing quality health services are met due to insufficient public funding, unequal access to services and the fact that patients often have to pay out of pocket. This has led to a vicious circle where underfunded services are unreliable, of poor quality, and not accountable to users.
To support the development of people-centered health systems, it is essential to establish appropriate funding mechanisms, including the amount of funding received, methods of funding to service providers and incentives created by payment mechanisms. Both financing and service delivery modalities should be addressed simultaneously. The shocks the economy and the health sector had to face during the last years have exposed the resilience of healthcare systems, highlighting the importance of strategic procurement to improve their efficiency. In this respect, Montás et al. developed a study for six EU countries and found that strategic buyers responded to seven health system shocks, with each country relying on the federal government for funding and response. Buyers often have limited, passive roles in contributing to a resilient health system, the need for strategic procurement in addressing health system challenges being very important (Montás et al., 2022).
The pandemic revealed the real situation of the health systems worldwide. There are significant adverse effects on the global health system, such phenomena having both economic and medical impact, with many negative consequences for the mental health of the population, consequences with significant effects even today (Vasile et al., 2022). Authors like Lupu and Tiganasu analyzed the efficiency of health systems in 31 European countries during the COVID-19 pandemic, using the methodological tools of DEA to calculate efficiency scores for all the health systems analyzed. Six major areas of influence were considered: healthcare, health status, population, economic, cultural/societal and governmental issues. The study showed that Western countries, especially Italy, Belgium, Spain and the UK, experienced high inefficiency in the first phase of the pandemic. However, Western states began to improve their medical systems during the relaxation phase and the second wave, proving their superiority, as proved in the current analysis. The study also found that the influencing factors varied for each stage: population age, population density, government effectiveness and education. The findings can help health policymakers compare good practices and develop national plans to better deal with future health crises (Lupu & Tiganasu, 2022).
The global situation characterizing the last years raised questions about global health policy, short-term financing solutions to help health systems cope with shocks. However, research on the relationship between pandemic resilience and long-term health policies is limited. In a recent research the authors Marginean and Orastean set out to investigate whether the countries of the EU were consistent in terms of financing the national health systems, and how prepared they were to face the shock. The analysis was carried out in 27 countries of the EU, divided according to the level of health expenditure, respectively high, medium and low. The results showed that countries in the high health spending cluster performed better, indicating that better financing could increase health system performance and resilience to future shocks (Marginean & Orastean, 2022).”
3) I would suggest the same caution and recommendation for the final section of the article. The authors highlight corruption in the health sector, which is not the subject of the model's analysis (p. 21, lines 764-766 and 758-759). If the authors feel that they are analysing the issue in question in a model, it may be necessary to add a "dummy" variable or similar to the model. Otherwise, I suggest shortening the conclusion and referring to the model result. Basically, the conclusions should be more general, as we know that it is a mathematical modelling tool and an approximation to real life.
Authors: Unfortunately, corruption in the health system is a reality in many countries and brings about negative effects like raising insufficient funds to purchase the adequate equipment, the medicines or attract the most qualified medical care personnel. Moreover, this phenomenon brings about low level of confidence and satisfaction among patients. However, since it is not the main topic addressed in this research, we have excluded from the conclusions the following parts: “In addition, corruption in the healthcare system seriously affects the quality of medical care, which leads to a low level of patient satisfaction.” and “At the same time, combating corruption in the healthcare system must be a priority, by implementing measures of transparency and financial responsibility towards the citizen.”
Suggested modifications have been made. Thank you.
After the corrections made, I believe that the article will be interesting, useful and credible for publication in the journal.
[1] Section 3 and its two subsections were initially integrated in the introduction. Since the introduction was unusually long (an issue also argued by one reviewer in his report), we decided to make it more concise and refer in a distinct part of the paper to the financing schemes of the health systems and the actual socio-economic background in the six EU countries selected for the proper analysis.
Reviewer 2 Report
Comments and Suggestions for Authors
I did not find any novelty in this paper. It wasn't easy to detect the original research questions. Notably, the research findings were expected, so I did not find a justification for the research.
Also, I am not convinced about the econometric model. The extremely high R-squared suggests the model has glitches. Also, there was no justification for using a particular econometric model.
Comments on the Quality of English Language
English is okay.
Author Response
I did not find any novelty in this paper.
Authors: In this paper, the authors have included as novelty elements the following ones: identifying a new research domain regarding the connection between economic development, health management and health financing performance; the realization of a comparison between different EU member states according to their economic development level in direct connection with the health financing performance; conceptualization of a new financial model of efficient allocation in an uncertain macroeconomic context; building of 7 public policies related to the study findings. Some of these elements are new and they were written in the Abstract and in Conclusions section.
It wasn't easy to detect the original research questions.
Authors: The authors propose the following research questions:
Q1. Is the health performance directly dependent on the allocation of government financial funds?
Q2. Does the public debt the performance of the financing health system?
Q3. Is performance in health systems dependent on the level of final consumption expenditure of households of a state?
Q4. Can be defined a financial allocation model in order to identify vulnerabilities of health systems in achieving a new level of performance?
These questions were introduced at the end of Introduction section.
Notably, the research findings were expected, so I did not find a justification for the research.
Authors: The justification of the research consists in identifying a new research domain regarding the connection between economic development, health management and health financing performance. As a result, the finding of this research are notable as well. We have introduced these information at the end of Introduction section.
Also, I am not convinced about the econometric model. The extremely high R-squared suggests the model has glitches. Also, there was no justification for using a particular econometric model.
Authors: In order to support our proposed model, we introduced and comment a new Table 3 regarding autocorrelations. According to the autocorrelation table (Table 3), the variables are independent of each other and the autocorrelation phenomenon does not exist (autocorrelation coefficients have values between -0.5 and 0.5, with two exceptions for the POP indicator).We justified the use of econometric model at the end of section 4.
Comments on the Quality of English Language
English is okay.
Reviewer 3 Report
Comments and Suggestions for Authors
This paper determines the performance aspects of health systems financing and efficient financing models in relation to the evolution of macroeconomic indicators. I think this topic is within the scope of the journal. I also think the topic is valuable to be researched. However, I think there are some unclear expressions in this paper. My concerns are as follows.
1. In the Abstract, the authors should draw attention to any research gaps and highlight the uniqueness of their study. A contribution part should be added to the abstract.
2. The introduction is very long. Authors should focus on important elements within the introduction.
3. The authors should discuss the empirical results with the existing field literature.
4. The part on the policy implications should be improved.
5. A flowchart of analyses should be included within the text.
Comments on the Quality of English Language
Minor editing of English language required
Author Response
This paper determines the performance aspects of health systems financing and efficient financing models in relation to the evolution of macroeconomic indicators. I think this topic is within the scope of the journal. I also think the topic is valuable to be researched. However, I think there are some unclear expressions in this paper. My concerns are as follows.
- In the Abstract, the authors should draw attention to any research gaps and highlight the uniqueness of their study. A contribution part should be added to the abstract.
Authors: We introduced all information in the Abstract according to your pertinent observations.
- The introduction is very long. Authors should focus on important elements within the introduction.
Authors: Suggested modifications have been made. We more than agree that the introduction was unusually long. We shortened it and kept only the general ideas meant to be presented in an introduction part to make it more concise. The financing mechanisms and the socio-economic background of the six selected countries, which initially were integrated in the introduction, are now to be found in a more appropriate form in section 3 entitled “The Health Financing Systems” and in its two subsections.
- The authors should discuss the empirical results with the existing field literature.
Authors: We did it at the end of Literature section. Our conclusions were followed by the comparison between the empirical results and the existing field literature.
- The part on the policy implications should be improved.
Authors: According to your opinion, we have introduced 7 public policy proposals at the end of the Conclusions section.
- A flowchart of analyses should be included within the text.
Authors: We did it in the new Figure 6.
Comments on the Quality of English Language
Minor editing of English language required
Authors: The English in the paper was reviewed.
Round 2
Reviewer 1 Report
Comments and Suggestions for Authors
No specific comment
Comments on the Quality of English Language
No specific comment
Reviewer 2 Report
Comments and Suggestions for Authors
There is sufficient improvement.